# Recent Advances in Electrochemical Aptasensors for Detection of Biomarkers

**DOI:** 10.3390/ph15080995

**Published:** 2022-08-12

**Authors:** Marjan Majdinasab, Jean Louis Marty

**Affiliations:** 1Department of Food Science & Technology, School of Agriculture, Shiraz University, Shiraz 71441-65186, Iran; 2Universite de Perpignan Via Domitia, 52 Avenue Paul Alduy, CEDEX 9, 66860 Perpignan, France

**Keywords:** aptamer, biosensor, aptasensor, electrochemical, biomarker

## Abstract

The early diagnosis of diseases is of great importance for the effective treatment of patients. Biomarkers are one of the most promising medical approaches in the diagnosis of diseases and their progress and facilitate reaching this goal. Among the many methods developed in the detection of biomarkers, aptamer-based biosensors (aptasensors) have shown great promise. Aptamers are promising diagnostic molecules with high sensitivity and selectivity, low-cost synthesis, easy modification, low toxicity, and high stability. Electrochemical aptasensors with high sensitivity and accuracy have attracted considerable attention in the field of biomarker detection. In this review, we will summarize recent advances in biomarker detection using electrochemical aptasensors. The principles of detection, sensitivity, selectivity, and other important factors in aptasensor performance are investigated. Finally, advantages and challenges of the developed aptasensors are discussed.

## 1. Introduction

In biomedical science, the early diagnosis of diseases increases the likelihood of successful treatment and the life quality of patients. The use of biomarkers is emerging as one of the most promising medical approaches for disease management [1]. The development of new methodologies for detecting and quantifying disease biomarkers with high accuracy and sensitivity has gained attention in the field of clinical analysis. In medical science, a biomarker is a measurable indicator of the presence of a particular disease, the severity of disease state or some other physiological state of an organism. Based on the World Health Organization (WHO) definition, the indicator can be chemical, physical, or biological, and the measurement may be functional, physiological, biochemical, cellular, or molecular [2]. Biomarkers can be classified according to various criteria. Based on their features, they can be divided into imaging biomarkers (e.g., magnetic resonance imaging (MRI)) or biomarkers with three subgroups: volatile like breath, body fluid, or biopsy biomarkers. Molecular biomarkers are non-imaging biomarkers with biophysical properties that can be measured in biological samples such as plasma, serum, cerebrospinal fluid, bronchoalveolar lavage, and biopsy. They include nucleic acid-based biomarkers such as gene mutations or polymorphisms and quantitative gene expression analysis, peptides, proteins, lipids metabolites, and other small molecules.

Biomarkers may also be classified (Figure 1) according to their application, such as for diagnostic purposes (e.g., cardiac troponin for the diagnosis of myocardial infarction), for disease staging (e.g., brain natriuretic peptide for congestive heart failure), for prognosis (e.g., cancer), for prediction (e.g., the presence of 12 single-nucleotide polymorphisms (SNPs) in a Han Chinese schizophrenic population), for susceptibility/risk (e.g., the recognition of apolipoprotein E (APOE) gene variations determining a higher risk of developing Alzheimer’s disease throughout life), for safety (e.g., the presence of the HLA-B*1502 allele as an indicator of an increased risk of serious and fatal skin reactions to carbamazepine) and for monitoring the clinical response to an intervention (e.g., HbAlc for anti-diabetic treatment). Another group of biomarkers named pharmacodynamics/response biomarkers includes those used in decision making in early drug development. For example, pharmacodynamic (PD) biomarkers are markers of certain pharmacological responses that are of special interest in dose-optimization studies [3].

A variety of human disease-specific biomarkers have been determined for the early diagnosis, monitoring, and prediction of different diseases including cancers and cardiovascular and metabolic disorders.

Given that early and accurate diagnosis of disease can significantly increase patient survival rates (to higher than 90%) and at the same time reduce health care costs and the length of stay in hospital [1]. The development of rapid methods for detecting and monitoring biomarkers is of particular importance. Conventional diagnostic strategies such as macroscopic imaging methods (e.g., mammography, colonoscopy, X-ray, and positron emission tomography), ultrasound, and tissue biopsies are widely used for disease diagnosis. However, these methods are not sensitive enough for early-stage diagnosis or for evaluating the patient’s response to treatment. These methods also require a complex operation, invasive biopsies, expert personnel, and long-term analysis [4]. Recently, tremendous efforts have been made in the field of biosensor technology for the rapid, sensitive, selective, cost-effective, portable, and point-of-care (POC) detection of specific biomarkers. Different kinds of biosensors have been developed for the early identification of biomarkers before symptoms appear [5,6,7,8]. Biosensors can be classified based on the type of biorecognition element (antibody, aptamer, nucleic acid probe, enzyme) and the transducer type (optical, electrochemical, thermal, piezoelectric) [9]. Electrochemical biosensors are promising analytical devices in the different fields of clinical diagnosis, food quality control, and environmental monitoring. The principle of detection with this type of biosensor is the change in electrical current resulting from the electrochemical interactions at the electrode surface [4]. Electrochemical biosensors convert biochemical statistics, such as analyte concentrations, to analytically useful signals such as current or voltage. Electrochemical biosensors may be divided into four main groups including amperometric, impedimetric, voltametric, and photoelectrochemical [10]. In electrochemical biosensors, the electrode is a key component that is applied as a solid support for the immobilization of biomolecules (e.g., enzyme, antibody, and nucleic acid) and electron movement [11]. Based on the electrode material, electrochemical biosensors can be classified into carbon-based (carbon and carbon nanomaterials including carbon nanotubes and graphene) and non-carbon-based (metal- and metal nanomaterial-based electrodes such as metallic and silica nanoparticles, nanowire, indium tin oxide, and organic materials) [11]. Electrochemical biosensors can be used for the detection of different kinds of biomolecules in the human body. They exhibit several advantages including the simultaneous detection of several analytes, high sensitivity and selectivity, robustness, real-time analysis, cost-effectiveness, no need for complicated and time-consuming sample preparation, and the possibility of using small amounts of samples for analysis. These features make electrochemical biosensors ideal for POC tests for monitoring biomarkers.

Like other types of biosensors, in electrochemical biosensors, different kinds of molecules including antibodies, enzymes, nucleic acid probes, and aptamers can be used as biorecognition elements. A biosensor designed using aptamers as the biorecognition element is called an aptasensor [12]. Aptamers are single strands of DNA or RNA molecules with 25–90 nucleotides developed by an in vitro selection called SELEX (systematic evolution of ligands by exponential enrichment). They can specifically bind to targets of different sizes from small molecules to whole cells by folding around the target and forming a unique three-dimensional structure [13]. Compared with antibodies, aptamers exhibit distinct advantages such as simple and fast chemical synthesis, easy modification with different functional groups and linkers, good stability, and low cost [14]. Aptamers can be termed “artificial antibodies” because they show similar or even better specificity to that of antibodies [14]. Such features have remarkably favored their application as highly selective biorecognition elements for application in various aspects of biotechnology including the development of biosensors. In recent years, a variety of aptamer-based biosensors (aptasensors) have been developed for detecting different kinds of analytes. In this regard, electrochemical aptasensors have attracted the attention of many researchers because of the benefits mentioned related to the aptamers and electrochemical biosensors. Although the scope of application of electrochemical aptasensors is in different fields such as food, environmental, and clinical analysis, due to the importance of clinical diagnosis, in this review article, we focus the application of this type of biosensor on monitoring and detection biomarkers.

## 2. Methodology

In this review, the recent progress in electrochemical aptasensors for the determination of biomarkers, their working principles, their strengths, and their weaknesses have been reviewed. In this study, more than 250 peer-review and review articles were investigated, and almost 90 articles were considered based on the year of publication (articles published in the last 5 years), the novelty and interestingness of the strategy used for diagnosis, and the diversity of the biomarkers diagnosed. Most of the published articles were in the field of cancer and cardiovascular disease biomarker detection, and as much as possible, examples of different types of these biomarkers were mentioned in this review to acquaint readers with the variety of biomarkers of diseases and their importance.

## 3. Electrochemical Aptasensors for Cancer Biomarkers

Cancer has become one of the most prevalent diseases in the world in recent years. This disease has more than 200 different types and affects more than 60 human organs. Early diagnosis is a main issue in cancer for enhancing the survivability rate of cancer patients. Unfortunately, many methods of diagnosing cancer in the early stages have failed which leads to a decrease in the survival rate of cancer patients [15]. More than 90% of cancer-related deaths are due to lack of early diagnosis and metastasis of the primary cancer tumor [16]. Therefore, the availability of suitable biomarkers and rapid, sensitive, and selective diagnostic methods allow for the screening of a large population in a short time and at low cost. In recent years, a wide range of biomarkers have been introduced to diagnose cancer. However, only a handful of them have been used clinically in the last 30 years [17].

Different antibody-based assays and immunosensors including enzyme-linked immunosorbent assay (ELISA) [18], chemiluminescent immunosensors [19], fluorescence and electrochemical immunosensors [20,21] and piezoelectric immunosensors [22] have been developed to detect cancer biomarkers. However, some limitations of antibodies including thermal instability and their low reproducibility in the mentioned techniques have prompted researchers to design aptamers to identify cancer biomarkers. In this regard, a variety of aptamers have been developed against specific cancer biomarkers and been employed in different types of biosensors. Owing to several advantages of electrochemical aptasensors, as mentioned earlier, in recent years, a variety of electrochemical aptasensors have been developed for the detection of cancer biomarkers. Electrochemical aptasensors for cancer biomarkers can be divided into three groups including: (1) those detecting protein tumor biomarkers such as PSA, carcinoembryonic antigen (CEA), and mucin 1 (MUC1); (2) electrochemical aptasensors for detecting circulating tumor cells (CTCs), such as EpCAM; and (3) electrochemical aptasensors for exosomes (i.e., oncoproteins, RNA, and DNA fragments) such as CD63 [23].

Breast cancer is a common cancer among women with high morbidity and mortality. MiRNA-21 is a strong biomarker for breast cancer that is highly expressed in cancer patients. The overexpression of MiRNA-21 in the serum is closely related to breast cancer and therefore can be used as a biomarker for the early diagnosis of breast cancer [24]. In this regard, an electrochemical aptasensor based on microgel nanocomposite was developed for the highly sensitive detection of MiRNA-21 as a biomarker. Microgels are microscale hydrogels whose tiny size makes them appropriate materials for the functionalization of solid surfaces including electrode surfaces [24]. The anti-fouling property of microgels prevents nonspecific reactions and interference in biosensor development. In this study, the microgel particles of gold nanoparticles (AuNPs) were wrapped in a mixture of acrylic acid (AAc) and *N*-isopropylacrylamide (NIPAm) for polymerization. Then, the gold electrode surface was modified with this porous network structure of AuNP@NIPAm–co–AAc microgels. The amino-modified DNA capture probe (complementary to the target RNA) was bound to the activated carboxyl groups of the particulate gel on the surface of the electrode and applied for MiRNA-21 detection using differential pulse voltammetry (DPV). The aptasensor was used for the detection of MiRNA-21 in the serum of patients without any sample preparation. The assay showed a limit of detection (LOD) of 1.35 aM with a linear range of 10 aM to 1 pM. The stability of the modified electrode was estimated to be more than a week, which is low. The reason for the low stability of the electrode could be related to the nature of the microgel that was used to modify the electrode.

Because multiple protein biomarkers can change abnormally in a particular disease, detecting only one protein biomarker can lead to the misdiagnosis of the disease [25]. In this regard, a sandwich-type electrochemical aptasensor was developed for the simultaneous detection of carcinoembryonic antigen (CEA) and cancer antigen 15-3 (CA 15-3), which are highly expressed in breast carcinomas [25]. A nanocomposite of AuNPs and 3D graphene hydrogel (AuNP/3DGH) was employed as a biosensing substrate for the surface modification of a glassy carbon electrode (GCE) (Figure 2). The role of 3DGH was to increase the electron transfer rate and electrical conductivity of the aptasensor by providing a wide available surface area. On the other hand, AuNPs on the surface of 3DGH not only played a role in accelerating electron transfer but also facilitated the aptamer immobilization with desirable bioactivity. The AuNP/3DGH-modified GCE was treated with 3—mercaptopropionic acid (MPA) in order to provide carboxylic groups and immobilization of 5′-amino-functionalized CEA aptamer I (CEAApI) and CA 15-3 aptamer (CAAp) on the surface of the electrode. Hemin and ferrocene played the role of redox probes for CEA and CA 15-3, respectively, and generated electrochemical signals for dual-biomarker diagnosis. Accordingly, with the immobilization of one or both specific aptamers of the biomarkers, three aptasensors were fabricated for the individual or simultaneous monitoring of CEA and CA 15-3. After the incubation of the biomarkers with a modified electrode, in order to reach a sandwich structure, a CEA aptamer II attached to AuNP hemin-graphene hybrid nanosheets (CEAApII/AuNP/HGNs) and a CA 15-3 aptamer attached to AuNP-modified ferrocene-graphene (CAAp/AuNP/Fc/G) bio-conjugates were placed on the surface of the modified electrode. DPV was measured to record the current responses of the hemin and ferrocene. The current responses were directly proportional to the concentrations of the biomarkers. The LODs of CEA and CA 15-3 were found to be 11.2 pg mL^−1^ and 11.2 × 10^−2^ U mL^−1^, respectively. The aptasensor was used for the simultaneous detection of biomarkers in the serum samples.

Prostate cancer is the most common cancer among men and the second-leading cause of cancer death in men. At the moment, the gold standard option for diagnosing prostate cancer is the measurement of prostate-specific antigen (PSA) in serum (as a prostate cancer biomarker), which is severely questioned due to the increasing number of unnecessary interventions in stunted or slow-growing tumors that increase patients’ general health and anxiety [26]. One the most popular methods for the diagnosis of prostate cancer is testing PSA levels in blood serum. However, this test show several disadvantages such as limited sensitivity of ~80%, low specificity of ~20% within “normal” ranges, and high false-positive rates, which leads to unnecessary biopsies and patient overtreatment [27]. Moreover, PSA can also be detected in other noncancerous prostate disorders such as benign prostatic hyperplasia (BPH), prostatitis, and trauma. Therefore, finding new biomarkers with the ability to accurately diagnosis early-stage prostate cancer is very desirable. Yan et al. (2022), used sarcosine as a promising biomarker in association with PSA to diagnose prostate cancer [27]. They developed an electrochemical aptasensor for the combined measurement of dual biomarkers that facilitated the robust detection of different states of prostate cancer, especially for early diagnosis. For the fabrication of this aptasensor, hierarchical MoS_2_ nanoflowers and spherical SiO_2_ nanoprobes were used as a functional interface and for signal amplification, respectively (Figure 3). Initially, the surface of the GCE was modified with MoS_2_ nanoflowers and applied as a functional biosensing interface. This trans-scale interface with natural hierarchical texture has the ability to anchor capture DNA as well as access and detect target analyte. Therefore, capture DNA consisting of an aptamer and ployC tail was deposited on the surface of nanoflower-like MoS_2_ structures via the stable absorption of polyC on MoS_2_ by van der Waals forces. In order to increase the assay sensitivity, spherical SiO_2_ nanoprobes were synthesized by capping with both electroactive redox indicator and carboxyl-modified DNA. Thus, two different electroactive labels (ferrocene, Fc, and methylene blue, MB) and two probes DNA (P_MB_ and P_Fc_) were employed to develop SiO_2_@MB@P_MB_ and SiO_2_@Fc@P_Fc_, respectively. Probe DNA specifically hybridized with the partial domain in the capture sequence (10 nt), which was sufficiently strong to make a duplex DNA as a recognition element but also was accessible for competitive displacement. In the presence of a target molecule, the SiO_2_ nanoprobe was dehybridized due to specific binding between target analyte and its aptamer. Thus, the electrochemical signal using square wave voltammetry (SWV) was significantly decreased. The LODs were 2.5 fg mL^−1^ and 14.4 fg mL^−1^ for PSA and sarcosine, respectively.

Among all types of cancer, lung cancer is the most common type of cancer, leading to about 25% of all cancer deaths [16]. Generally, the lung cancer biomarkers are classified into two groups including DNA/genetic-based biomarkers (e.g., RAR-β mRNA, COX2, DAPK, RASSFIA, IL-8 mRNA, PRCS3, FHIT, K-ras mutant, p53 mutant) and protein-based biomarkers (e.g., CEA, CYFRA 21-1, TPA, tumor M2-pyruvate kinase, haptoglobin-R 2, APOA1, KLKB1, ProGRP) [16,28]. In the early stages of the disease, only small levels of biomarkers are present in the cancerous cells and thus in the body fluids [16]. Therefore, the development of credible and sensitive detection tests is of great importance. In this regard, an electrochemical aptasensor using LC-18 aptamer was developed for the detection of lung cancer biomarkers in real blood plasma [29]. The LC-18 aptamer is a highly specific aptamer with the ability to identify lung cancer-related proteins and cells. The aptasensor was fabricated using the immobilization of thiolated aptamer on the surfaces of gold disc electrodes. In order to inhibit any non-specific interactions, the uncovered surfaces of modified electrodes were blocked using blocking thiolated oligonucleotides. CV and non-Faradic EIS were used for the characterization of the aptasensor and when analyzing samples to detect analyte. The main disadvantage of this research was that sensitivity, selectivity, and stability of the aptasensor during storage were not investigated.

Ovarian cancer is one of the most common types of cancer in women worldwide and can lead to death. One of the main reasons for the high mortality rate due to ovarian cancer is the lack of a suitable biomarker for the early diagnosis of this cancer [30]. Moreover, due to the late onset of symptoms of ovarian cancer, most patients are diagnosed late. Carbohydrate antigen 125 (CA-125) and mucin 16 (MUC16) is a glycoprotein found in ovarian cancer cells that can be used in the diagnosis of ovarian cancer [31]. This biomarker is found in normal values (0–35 U mL^−1^) in the blood of healthy women. Values above this range in the bloodstream indicate the progression of ovarian cancer [30]. Therefore, the detection of CA-125 biomarker can be used to monitor ovarian cancer patients during and after treatment. In this regard, a label-free electrochemical aptasensor using a nickel hexacyanoferrate (NiHCF) nanocube as the in situ signal probe and polydopamine-functionalized graphene (PDA@Gr) as the substrate was developed for the detection of CA-125 [32]. NiHCF nanomaterials as one of the polynuclear transition metal hexacyanoferrates (MHCF) reveal a reversible and reproducible redox signaling response and high stability on the electrode surface. However, they suffer from low conductivity. Therefore, their combination with other nanomaterials or their application as nanocomposites can overcome this undesirable property. In this study, NiHCF/PDA@Gr nanocomposite was used for the surface modification of GCE. Then, amino-modified CA-125 aptamers were covalently bound with polydopamine by the Michael addition reaction. CA-125 was detected by measuring DPV. Upon the binding of CA125 to the immobilized aptamer on the electrode surface, an insulating layer was formed that hindered the electron transfer and resulted in the reduction of the NiHCF signal. The aptasensor exhibited a low LOD of 0.076 pg mL^−1^ with a linear range of 0.10 pg mL^−1^ to 1.0 μg mL^−1^. The aptasensor showed good selectivity and excellent reproducibility and stability (3 weeks). The applicability of the aptasensor was evaluated based on the detection of CA-125 in human serum samples.

A summary of the recent studies of electrochemical aptasensors for the detection of cancer biomarkers is provided in Table 1.

## 4. Electrochemical Aptasensors for Cardiac Biomarkers

Cardiovascular diseases (CVDs) are a main cause of deaths worldwide, especially in developing countries [33]. Therefore, the early and rapid diagnosis of CVDs is of special importance in the appropriate and timely treatment of patients and the reduction of CVD mortality rates. Generally, CVDs are a group of diseases that affect heart or blood vessels. They are associated with various symptoms including myocardial rupture, arrhythmia, congestive heart failure, cardiogenic shock, and pericarditis [33]. A group of biomarkers including cardiac troponin (cTn) I or T, lipoprotein-bound phospholipase, lipoprotein-bound phospholipase, low-density lipoprotein, c-reactive proteins (CRP), myoglobin, interleukin-6, interleukin-1, tumor necrosis factor α, and myeloperoxidase have been identified for CVDs and can be used for the early diagnosis of this group of diseases [33,34]. Among these, CRP and cTn are the two most important biomarkers. The high specificity of these two biomarkers along with their long persistence in the blood has made them good biomarkers in the diagnosis of acute myocardial infarction (AMI) [34]. In this regard, Chen et al. (2022) developed a highly sensitive electrochemical aptasensor based on the clustered regularly interspaced short palindromic repeat (CRISPR)/Cas12a system for the determinarion of cTnI [35]. Cas12a (CRISPR-associated protein 12a) as a part of the CRISPR systems is an RNA-guided endonuclease with the role of effector protein in CRISPR-Cas systems. Cas12a employs the crRNAs as a guide for binding to complementary dsDNA or ssDNA sequences, leading to its collateral cleavage activity toward the nonspecific ssDNA reporter being triggered [35]. In this study, the biotin-modified cTnI aptamer and the (nucleic acid probe 2) P2 were first hybridized and then bound to the streptavidin-modified magnetic nanoparticles (MNPs). Upon adding cTnI, the cTnI aptamer in the duplex specifically bound to cTnI resulted in the dissociation of the P2. After magnetic separation, the suspension was combined with cas12a/crRNA, and the modified gold electrode was dipped in this mixture. The released P2 was bound to cas12a/crRNA, and the trans-cleavage activity of Cas12a/crRNA was switched on. Thus, the methylene blue-functionalized DNA probe 1 (P1) on the electrode surface was cleaved, resulting in the decrease of the electrochemical signal in DPV measurement. The assay was highly sensitive, with a LOD of 10 pg mL^−1^ and a linear range of 100–50,000 pg mL^−1^. The fabricated aptasensor was successfully used for the determination of cTnI in human serum.

In order to increase the sensitivity of cTnI detection, Villalonga et al. developed a novel sandwich-type aptasensor using a new carboxyethylsilanetriol-modified graphene oxide (GO) as transduction element [36]. Carboxylic acid groups could act as points for the immobilization of the bioreceptor. On the other hand, the covalent modification of GO with chloroacetic acid and 6-aminohexanoic acid increased the load capacity of the bioreceptors. After the modification of screen-printed carbon electrodes (SPCEs) with carboxyethylsilanetriol-modified GO, the amino-functionalized anti-cTnI-specific ssDNA aptamers were immobilized onto modified SPCE through covalent attachment with -COOH residues. The sensing strategy was based on the specific detection of cTnI by the aptamer and the subsequent assembly of sandwich-type architecture with a new peroxidase-anti cTnI conjugate as signaling element, resulting in the amperometric detection of the cTnI biomarker following incubation with a mixture of hydroquinone and H_2_O_2_. The aptasensor showed an LOD of 0.6 pg mL^−1^ a linear range of 1.0 pg mL^−1^ to 1.0 µg mL^−1^. It was applied for the detection of cTnI in serum samples.

In another study, a label-free electrochemical aptasensor was developed for the detection of CRP [37]. First, GO was modified with poly deep eutectic solvent (PDES/GO) (Figure 4). Then, PDES/GO was coated with the AuNPs. CRP DNA aptamers were deposited on the surfaces of Au NP-modified electrodes through gold–sulfur affinity. DESs are nonflammable, thermally stable, biodegradable, relatively less toxic, nonvolatile, and cheap to produce. The functionalization of graphene with PDESs results in stability, uniform dispersal, high functionality, high surface charge density, and control of the AuNPs’ size [37]. AuNPs act as signal amplifiers. In the presence of CRP and its specific binding to the aptamers on the surfaces of electrodes, the resistance (ΔR_ct_) was increased due to providing an insulating layer on the electrode surface. The designed aptasensor showed a LOD of 0.0003 ng mL^−1^ with a linear range of 0.001–50 ng mL^−1^. It was applied for the detection of CRP in serum samples with a good recovery.

Inspired by enzyme-inorganic hybrid nanomaterials that play an important role in signal output and amplification in bioassays, but suffer from inactivation, aptamer–inorganic hybrid nanomaterials were synthesized and applied for the electrochemical determination of CRP biomarker [38]. In this study, Mn_3_(PO_4_)_2_/CRP aptamer nanosheets with both functions of biorecognition and signal amplification were fabricated by one-pot biomineralization. First, GCE was modified with polydopamine/AuNPs (PDA/AuNPs), which on the one hand provided active sites for the immobilization of primary antibody and on the other hand increased the electron transfer on the electrode surfaces. After antibody immobilization on the modified electrode surface, the electrode was incubated with CRP and the Mn_3_(PO_4_)_2_/CRP aptamer solution. Then, a Na_2_MoO_4_ solution was dropped onto the modified electrodes. The specific binding between CRP and its aptamer immobilized in nanosheets was followed by introducing molybdate to form the molybdophosphate precipitates and production of the redox signals. SWV was used for electrochemical measurements. The electron-transfer resistance (Ret) was increased progressively after applying antibody, CRP and Mn_3_(PO_4_)_2_/aptamer onto the electrode. The LOD obtained was 0.37 pg mL^−1^, with a linear range of 1–1000 pg mL^−1^. The aptasensor was successfully applied for the detection of CRP in serum samples.

A summary of the recent studies of electrochemical aptasensors for the detection of cardiac biomarkers are provided in Table 2.

## 5. Electrochemical Aptasensors for Alzheimer’s Disease Biomarkers

Alzheimer’s disease (AD) is the most widespread neurodegenerative illness, manifested as memory loss, cognitive dysfunction, speech loss, and personality change that strongly affect personal, social, and economic functions [39]. Based on the WHO, AD is currently the main cause of disability for elderly persons worldwide [40]. The disease is usually diagnosed late. As many as 75% of patients are unaware of their disease [40]. Unfortunately, no effective drug has been identified for the radical treatment of Alzheimer’s disease. Therefore, the early diagnosis of this disease is necessary in order to treat and prevent its progression. Cerebrospinal fluid (CSF) is a clear and colorless fluid that supplies nutrients to the brain and helps eliminate waste products of the brain’s metabolism. AD is basically recognized via two different proteins in CSF including amyloid-β (Aβ) and tau protein, which respectively overaccumulate outside and inside neurons [40,41,42]. These two proteins are the most important biomarkers for the diagnosis of AD. Another important biomarker is apolipoprotein E. Furthermore, several other biomarkers have been recently recognized as nonspecific biomarkers including dopamine, interleukin-6, neurofilament light (NfL), flotillin, alpha-1 antitrypsin (AAT), amyloid precursor protein (APP), amyloid-β-derived diffusible ligands (ADDLs), thiamine pyrophosphate, β-site APP-cleaving enzyme (BACE1), adenosine triphosphate (ATP), α-synuclein, microRNAs (miRNAs), glycated albumin, and Alzheimer-associated neuronal thread protein (NTP) [39,40].

Using amyloid-β as a biomarker, an electrochemical aptasensor based on a composite of superhydrophobic carbon fiber paper (CFP) and AuPt alloy nanoparticles (CFP/AuPt) was developed [43]. The high specific surface of AuPt alloy nanoparticles enhanced the binding sites and the electron transfer speed, which significantly increased the assay sensitivity. On the other hand, the hydrophobic surface of CFP/AuPt increased the resisting nonspecific adsorption efficiency of the biosensor, which resulted in reduced interference in the complex biological samples. In this system, the thiol-modified amyloid-β DNA aptamers were immobilized on the CFP/AuPt electrode using a self-assembling method. In the presence of amyloid-β target and the complex formation of aptamer-amyloid-β, the electron transfer on the electrode surface was inhibited. DPV was measured for the determination of amyloid-β. Under optimal conditions, the biosensor showed a LOD of 0.16 pg mL^−1^ with a linear range of 0.5–10,000 pg mL^−1^. The aptasensor was successfully applied for the detection of amyloid-β in serum samples.

Another important biomarker for AD detection is tau protein, which is an intrinsically unfolded protein and plays an important role in microtubule filament stabilization, the facilitation of neuronal transport, and keeping cell integrity [39]. In this regard, Hun et al. developed an enzyme-linked aptamer photoelectrochemical aptasensor based on AuNPs/MoSe_2_ nanosheets as the sensing platform for the detection of Tau-381 protein [44]. A carbon paste electrode (CPE) was modified with AuNPs/MoSe_2_ nanosheets, which enhanced the aptamer binding sites and electron transfer rate (Figure 5). Then, thiolated Tau-381 protein aptamer was immobilized on the surface of the modified electrode. On the other hand, Tau-381 antibody and the protein G labeled with alkaline phosphatase (protein G/AP) were used for generating photoelectrochemical signal. In the presence of Tau-381 protein, a sandwich complex was formed between the aptamer, the Tau-381 protein, the antibody, and G/AP. Upon adding ascorbic acid 2-phosphate (AAP) and catalyzing with AP, ascorbic acid was produced as an electron donor. As a result, a strong photocurrent response was produced. The LOD was 0.3 fM with a linear range of 0.5 fM to 1.0 nM.

Thrombin is a serine protease with the ability to catalyze the conversion of fibrinogen to fibrin and blood coagulation. The overexpression of thrombin can contribute to some blood diseases including leukemia, cardiovascular disease, liver disease, and Alzheimer’s disease [45]. Therefore, it can be also considered a biomarker of AD. Different kinds of biosensors such as colorimetric, fluorescence, and electrochemical have been developed for the detection of thrombin. For example, Sun et al. (2022) fabricated an electrochemical aptasensor that operated as a hyaluronic acid functionalized polydopamine for the determination of thrombin in human serum [45]. Hyaluronic acid with hydroxyl groups was immobilized on the surface of polydopamine (as an excellent coating material) modified GCE using the linkage of 6-mercapto-1-hexanol to provide antifouling action. Moreover, the hyaluronic acid supplied a suitable substrate for the immobilization of thrombin aptamers through the formation of the amide bond between the carboxyl group of hyaluronic acid and the amino group of the aptamer. Thrombin was measured by DPV, such that DPV peak current decreased by increasing the concentration of thrombin. Due to the presence of hyaluronic acid in the sensing interface, the aptasensor was able to detect thrombin in the serum samples with insignificant effects of nonspecific adsorption. The aptasensor showed a high sensitivity, with a LOD of 0.03 pM and a linear range of 0.1 pM to 1.0 nM.

A summary of the recent studies of electrochemical aptasensors for the detection of Alzheimer’s disease biomarkers are provided in Table 3.

## 6. Electrochemical Aptasensors for Multiple Sclerosis (MS) Biomarkers

Multiple sclerosis (MS) is the most common autoimmune demyelinating disease. In MS, the myelin sheath, which is the insulation that covers the nerve cells, is damaged. This damage results in a disruption in the transmission of nerve messages that is accompanied with a range of symptoms including double vision, blurred vision, blindness in one eye, problems with arm or leg movement, muscle weakness, and difficulty with sensation or coordination [46]. The most common methods of MS detection are magnetic resonance imaging (MRI) and CSF analysis, which are expensive and time-consuming and carry adverse effects [47,48]. The detection of specific biomarkers can be effective in the early and rapid diagnosis of MS. Different cell-level MS biomarkers are found in CSF. According to the literatures, three different groups of biomarkers including laboratory biomarkers, commonly body fluids (i.e., oligoclonal bands (OCB), measles-rubella-zoster (MRZ), kappa-free light chains (kFLCs), CXCL13, neurofilament light chain (NfL), autoantibodies to myelin basic protein (anti-MBP) and vitamin D); markers obtained from imaging technologies (i.e., MRI); and genetic–immunogenetic markers, which are related to genetics and immunogenetics (i.e., circulating MicroRNAs (miRNAs)) have been determined for MS disease [47,48]. One of the main biomarkers for MS diagnosis is miR-155. MicroRNAs are small non-coded RNAs with approximately 19 to 24 nucleotides. They are abnormally expressed in various diseases and have been determined as predictive biomarkers for the early diagnosis of diseases [46]. Accordingly, Shariati et al. (2022) developed an electrochemical aptasensor for the detection of miR-155 in human serum [46]. In this study, the surface of a graphite sheet (GS) was coated with a nanocomposite of single-walled carbon nanotubes (SWCNTs) and polypyrrole (PPY) in order to enhance the active surface area, accelerate the electron transfer, increase the electrical conductivity, and decrease overvoltage (Figure 6). Then, amine-modified aptamer specific to the miR-155 was bound to carboxylic groups of SWCNTs through amide bonding. DPV was measured for target detection. The complex formation between miR-155 and its specific aptamer resulted in reducing the peak current height, which was related to the decrease in conductivity on the electrode surface. The aptasensor showed a LOD of 10 aM with the linear range of 10 aM to µM.

Despite the recent progress in the identification of various biomarkers using aptamers and electrochemical aptasensors, not many studies have been conducted in the field of detecting specific biomarkers of MS disease using aptasensors, especially electrochemical aptasensors. Most of the studies in the field of electrochemical aptasensors in the diagnosis of this disease have investigated and determined general biomarkers such as dopamine, interleukin-17 (IL-17), and epinephrine. Interleukin-17 receptor A (IL-17RA) is a valuable biomarker associated with numerous autoimmune diseases including psoriasis, inflammatory bowel disease, asthma, type 1 diabetes, rheumatoid arthritis, and MS. Jo et al. (2016) developed a sensitive electrochemical aptasensor for the detection of IL-17RA [49]. To fabricate the aptasensor, thiolated aptamer specific to IL-17RA was immobilized on the surface of a gold electrode modified with AuNPs through gold–thiol interaction. In the presence of IL-17RA target and complex formation of an IL-17RA- ptamer, the resistance increased on the surface of the electrode. EIS was used to measure IL-17RA concentration in serum samples. The LOD and linear range were obtained 2.13 pg mL^−1^ and 10–10,000 pg mL^−1^, respectively.

A summary of the recent studies of electrochemical aptasensors for the detection of MS biomarkers are provided in Table 4.

## 7. Electrochemical Aptasensors for Malaria Biomarkers

Malaria is a serious vector-borne disease caused by *Plasmodium* parasites. This disease is one of the oldest infections and, despite many efforts, is still prevalent in tropical and developing countries and leads to the death of hundreds of thousands of infected people every year [50].

Therefore, the WHO seriously advises its strategy of “test, treat, and track” in order to improve the quality of treatment and monitor malaria outbreaks. Moreover, the WHO advocate that all suspected malaria cases should be confirmed by diagnostic testing before treatment is ordered to inhibit parasite resistance [50]. *Plasmodium falciparum* and *Plasmodium vivax* are the two main malaria parasite species [51]. For the diagnosis of malaria, different marker proteins including lactate dehydrogenase (LDH), histidine rich protein-II (HRP-II), aldolase, and glutamate dehydrogenase (GDH) have been evaluated [52]. *Plasmodium falciparum* lactate dehydrogenase (PfLDH) has been recognized as an important malaria biomarker since it can be detected in all blood stages and is overexpressed n the metabolism of *Plasmodium* parasites [51]. A DNA aptamer (ssDNA) has been developed for the specific detection of PfLDH. Accordingly, Figueroa-Miranda et al. (2020) developed an electrochemical aptasensor for the detection of PfLDH in human serum without complicated sample pretreatments [51]. In this study, a gold rod electrode was coated with a PfLDH aptamer and polyethylene glycol (PEG) stepwise. PEG was used as an antifouling agent in order to reduce the nonspecific adsorption of other proteins or cells in a serum sample. Upon the target PfLDH binding to its specific aptamer, charge transfer resistance increased. EIS was used to determine PfLDH concentrations. The LOD and linear range obtained were 1.49 pM and 4.5 pM–100.0 nM, respectively, in whole human serum.

Another important biomarker for malaria diagnosis is PfGDH. It has been detected as a powerful biomarker due to its obvious distinctions from the host GDH in terms of structure, sequence, location, and enzyme kinetics [52]. Singh et al. developed a label-free capacitive aptasensor for the detection of PfGDH in serum samples (Figure 7) [52]. The aptasensor was fabricated through the immobilization of thiolated ssDNA aptamer on the surface of the gold disc electrode followed by backfilling/blocking with 6-mercapto 1-hexanol (MCH). A non-Faradaic EIS-based technique was used to produce the response. In the presence of PfLDH and binding to its specific aptamer, charge transfer resistance increased. The aptasensor showed a wide linear range from 100 fM to 100 nM with a LOD of 0.77 pM in serum samples.

In another study, HRP-II was detected as a malaria biomarker using an EIS aptasensor [53]. For the fabrication of the aptasensor, the amine-functionalized HRP-II aptamer was immobilized on the surface of the gold electrode using a self-assembled monolayer of dithio-bis(succinimidyl) propionate. The complex formation of the aptamer and HRP-II protein reduced the exposed negative charge of the aptamer layers that facilitated the charge transfer on the electrode and thus, reduced the R_CT_. The LOD obtained was 3.15 pM, with a linear range of 1–500 pM. 

A summary of the recent studies of electrochemical aptasensors for the detection of malaria biomarkers are provided in Table 5.

## 8. Electrochemical Aptasensors for Diabetes Biomarkers

Diabetes mellitus, commonly known as diabetes, is a chronic disorder specified by prolonged high blood sugar levels (hyperglycemia) [54]. The symptoms include frequent urination, increased thirst, and increased appetite. However, untreated cases can lead to much worse health complications. The major complications of diabetes are “microvascular (retinopathy, nephropathy, neuropathy) and macrovascular (ischaemic heart disease, stroke, peripheral vascular disease) endpoints” [54]. Two main forms of diabetes include type 1 and type 2 according to the majority (>85%) of total diabetes prevalence. However, diabetes can be also evident during pregnancy and under some conditions including genetic disorders, drug or chemical toxicity, endocrinopathies, insulin receptor disorders, and in association with pancreatic exocrine disease [54,55].

The conventional detection method of diabetes is based on glucose measurement such as impaired fasting glucose (IFG) and impaired glucose tolerance (IGT), which can lead to diagnosis errors [56]. Recently, glycated hemoglobin (HbA1c) has been considered an indicator of blood glucose levels in patients in the last 60 to 90 days [56]. Therefore, it can be an excellent biomarker and a gold standard for the continuous monitoring of glucose. HbA1c is a glycated protein with high stability that is produced by the non-enzymatic reaction of glucose with human hemoglobin (Hb) β-chain *N*-terminal valine in serum. The HbA1c level (%) reflects the time-averaged blood glucose level over a period of two to three months [57]. In 2010, a diagnostic threshold of ≥6.5% HbA1c was recommended as one of the criteria for diabetes by the American Diabetes Association [57]. A variety of analytical approaches have been used for the detection of HbA1c level in human blood including capillary electrophoresis, immunoassays, boronate affinity chromatography, enzymatic assays, and different kinds of biosensors. Eissa et al. (2017) developed a label-free electrochemical aptasensor array for the detection of total hemoglobin (tHb) and HbA1c in human whole blood [58]. Thiolated aptamers specific to HbA1c and tHb were immobilized on the surfaces of AuNP modified-array screen-printed carbon electrodes and employed for the label-free detection of HbA1c and tHb using SWV. After the complex formation of the analyte–aptamer, a decrease in the peak current was observed due to the blocking effect of these bulky proteins. The LODs obtained were 0.2 and 0.34 ng mL^−1^ for HbA1c and tHb, respectively. The linear range of aptasensor was estimated to be 100 pg mL^−1^ to 10 µg mL^−1^. The aptasensor was used for the detection of HbA1c % in human whole blood without any pre-treatment.

Glycated human serum albumin (GHSA) is a medium-term glycemic control biomarker of diabetes that can be employed as an alternative to or together with HbA1c. Accordingly, Bunyarataphan et al. (2019) developed an electrochemical aptasensor for the detection of GHSA using two DNA aptamers that specifically bound to GHSA and HSA [59]. For the fabrication of the aptasensor, biotinylated aptamers were immobilized on the surface of SPCE modified with streptavidin (STR). SWV was used to measure the binding of the target proteins to their specific surfaces. Upon the addition of GHSA and the complex formation of GHSA/aptamer, the electron transfer rate was hindered, and the electrochemical signal was reduced. The aptasensor showed LODs of 3 ng mL^−1^ and 0.2 µg mL^−1^ for GHSA and HAS, respectively. The linear ranges obtained were 2 × 10^−6^–16 mg mL^−1^ for GHSA and 5 × 10^−5^–100 mg mL^−1^ for HSA. The aptasensor was used for the detection of GHSA and HSA in clinical plasma samples.

Serpin A12 (visceral adipose tissue-derived serpin) is a novel biomarker of type 2 diabetes. It is produced by visceral and subcutaneous adipose tissues. Serpin A12 is expressed in the mRNA or protein in some tissues including liver, pancreas, and skin. It acts as an insulin sensitizer with an anti-inflammatory effect [60]. Previous studies have shown that the concentration of this compound increases in obesity or insulin resistance conditions [61]. Therefore, it can be considered a suitable biomarker for type 2 diabetes. Salek Maghsoudi et al. (2020) developed an electrochemical aptasensor for the detection of serpin A12 [60]. In this study, SPCE was modified with flower-like gold microstructure (FLGM) in order to increase electrochemical performance. Then, thiolated aptamer specific to serpin A12 was immobilized on the surface of modified electrode. DPV was measured for the determination of different concentrations of serpin A12. DPV peak increased with increasing concentrations of serpin A12. The aptasensor showed LODs of 0.02 and 0.031 ng mL^−1^ in the buffer and plasma samples, respectively, with a linear range of 0.039–10 ng mL^−1^.

Diabetic retinopathy (DR) is a complication of diabetes that affects the eye. It can lead to vision loss and blindness if left undiagnosed and untreated [62]. Therefore, the early diagnosis and treatment of DR can effectively inhibit severe irreversible damage. Vascular endothelial growth factor (VEGF) is an important tear biomarker for DR, such that an increased concentration of VEGF is highly associated with DR [63]. In this regard, Mei et al. (2021) fabricated a reusable electrochemical aptasensor for the diagnosis of DR through the VEGF tear biomarker [63]. In this study, they employed hybridization chain reaction (HCR) and CeO_2_ NPs as cascade signal amplification methods to develop the VEGF aptasensor. As shown in Figure 8, in the first part hairpin, cDNA was mixed with sample and trapped with VEGF. In the second part, the aDNA was immobilized on the surface of a gold electrode through the Au–S bond. The free cDNA was hybridized with aDNA. The report probes H1 and H2 functionalized with CeO_2_ NPs were added to HCR, under the initiation of active aDNA, to open the hairpin structure. Due to the binding of VEGF to cDNA, excess aDNA triggered the HCR reaction. The first analysis approach for the detection of VEGF was to use CeO_2_ NPs to catalyze H_2_O_2_ to produce electrochemical signals; this was quantified by DPV. The concentration of VEGF was positively related to the HCR products. The developed biosensor showed a highly sensitive detection of VEGF down to 0.27 fg mL^−1^, with a linear range from 1 fg mL^−1^ to 0.1 ng mL^−1^. Next, hairpin rDNA was replaced with cDNA on the electrode by strand displacement reaction (SDR) to activate sites of aDNA. The aptasensor performed a second repeated assembly to produce a second signal on SDR. The ratio of the former and latter signals was employed to perform a second analysis method for determining VEGF. The second analysis approach showed a LOD as low as 7.39 fg mL^−1^, with linear ranges from 10 fg mL^−1^ to 0.1 ng mL^−1^.

A summary of the recent studies of electrochemical aptasensors for the detection of diabetes biomarkers are provided in Table 6.

## 9. Electrochemical Aptasensors for Other Biomarkers and Microbial Pathogens

Electrochemical aptasensors can also be used for the detection of biomarkers of other types of diseases and pathogenic microorganisms (e.g., viral and bacterial agents). For instance, Naseri et al. developed an electrochemical aptasensor for the detection of lactoferrin as a biomarker of urinary tract infection [64]. In this study, the surface of the screen-printed gold electrode was modified with a multivalent aptamer, with multivalent binding ability, in order to improve the binding affinity of the aptamer to human lactoferrin. Then, it was used for the label-free detection of lactoferrin in buffer solution and spiked urine samples. DPV was used for the determination of lactoferrin. By increasing the lactoferrin concentration, oxidation potential was negatively shifted, which could be attributed to facilitating charge transfer in the presence of lactoferrin. The aptasensor showed a LOD of 0.9 ng mL^−1^ and a wide linear range from 10 to 1300 ng mL^−1^. In addition to high sensitivity, the new platform exhibited high selectivity, excellent reproducibility, and a simple sensing procedure.

Vitamin D and 25-hydroxy vitamin D3 (25OHVD3) play a crucial role in human health and its deficiency has been shown in many diseases such as osteoporosis, mental illness, and coronavirus disease 2019. Furthermore, 25OHVD3 is associated with various diseases such as coronary heart disease and cancer. Therefore, the development of sensitive, fast, and accurate methods of determining 25OHVD3 is crucial. Yin et al. compared two aptamer platforms to develop an electrochemical aptasensor for the detection of 25OHVD3 [65]. The oligonucleotide aptamers were immobilized on a gold surface using a disulfide linker and monitored using different electrochemical methods including cyclic voltammetry (CV), SWV, and EIS. EIS was used for the determination of different concentrations of 25OHVD3. R_CT_ progressively increased when the target compound bound with the aptamer on the gold surface. The results showed that aptamer VDBA14, possessing a 5′-terminal disulfide, recognizes the vitamin target with a LOD of 0.085 nM and a linear range of 1–1000 nM. The fabricated aptasensor was used for detecting 25OHVD3 in human serum samples with high sensitivity and selectivity.

Considering that the cause of many diseases, such as smallpox, maltose fever, common cold, and gastric ulcer, are pathogenic microorganisms (viruses and bacteria), their identification can be a good indicator to ensure the type of disease. Therefore, the sensitive and selective detection of pathogens in complex actual samples is very important. The importance of this issue was seen to a greater extent during the coronavirus disease 2019 (COVID-19) pandemic amid the need for its rapid and accurate diagnosis. During this pandemic, many studies were conducted to develop different kinds of detection kits and biosensors. The results of these studies range from the production of rapid diagnostic tests for home use to the production of accurate and sensitive electrochemical biosensors. In this regard, Rahmati et al. developed an electrochemical aptasensor for the label-free detection of SARS-CoV-2 spike glycoprotein base on the composite of Cu(OH)_2_ nanorods arrays [66]. In this research, the surface of SPCE was coated with Cu(OH)_2_ nanorods. This structure provided high surface area in the small space of the SPCE surface. In the presence of SARS-CoV-2 spike glycoprotein, a decrease in Cu(OH)_2_ nanorods-associated peak current was observed that could have been due to the target-aptamer complex formation and thus blocking the electron transfer of Cu(OH)_2_ nanorods on the surface of SPCE. SWV was used for the determination of SARS-CoV-2 spike glycoprotein. The aptasensor showed a LOD of 0.03 fg mL^−1^ with a linear range from 0.1 fg mL^−1^ to 1.2 µg mL^−1^. The assay was performed within 15 min. High selectivity and good reproducibility were other characteristics of the developed platform.

*Helicobacter pylori* is a Gram-negative, spiral-shaped, fastidious, and microaerophilic bacterium. It is the leading cause of chronic gastritis as well as duodenal and gastric ulcers. The detection of *H. pylori* infection is important for the choice of treatment and the success of eradication. Therefore, the sensitive and selective detection of the *H. pylori* is crucial. In this regard, Sarabaegi et al. fabricated an ultrasensitive electrochemical aptasensor for the rapid detection of *H. pylori* [67]. In this study, the surface of GCE was modified with an electrospun nano-sized bimetallic NiCo-metal-organic frameworks nanostructure (NiCo-MOF@C) as platform. Due to the hydrogen bonds, large amounts of aptamer strands could be bound to the NiCo-MOF@C, resulting in a low LOD of 1 CFU mL^−1^ and linearity of 10^1^–10^7^ CFU mL^−1^. EIS was used for the detection of *H. pylori.* After the incubation of the *H. pylori* on the aptamer/NiCo-MOF@C/GCE, the R_ct_ increased. The developed aptasensor showed high selectivity and reproducibility. The aptasensor was successfully applied for the detection of *H. pylori* in serum bloods diluted with phosphate buffer.

## 10. The Performance of Electrochemical Aptasensors for Biomarker Detection

In recent years, different kinds of electrochemical aptasensors have been developed for the detection of a variety of biomarkers. The performance of these aptasensors depends on different parameters such as electrode type, the manner of electrode modification, using or not using nanomaterials in the sensor-fabricating process, the use of signal amplification strategies, and electrochemical method (e.g., voltammetry, amperometry, impedance). Moreover, the structural characteristics of the aptamer including type (DNA or RNA), sequence length, functionalization, and affinity to the target play an important role in aptasensor performance. These factors can greatly affect the sensitivity, selectivity, stability during storage, and reproducibility of an aptasensor. As presented in the tables, most of the developed aptasensors show high sensitivity due to the utilization of nanomaterials to modify the electrode surfaces, which not only increases surface area for the immobilization of a large number of bioreceptor molecules (e.g., aptamers) but also increases electrical conductivity, resulting in the enhanced detection signal. AuNPs are one of the most widely used nanomaterials in the development of electrochemical aptasensors due to their beneficial features such as easy synthesis and easy functionalization with thiolated aptamer through the affinity of the thiol group to gold. On the other hand, the use of aptamer as a biorecognition element has provided high selectivity in diagnosing the desired biomarker. Most apatasensors for biomarker detection have used DNA aptamer rather than RNA aptamer due to several advantages of DNA aptamer such as higher stability, easy functionalization, lower cost, and a simple selection procedure [68].

Another important factor in the efficiency of an aptasensor is its stability during storage. From the results shown in the tables, it can be concluded that most developed aptasensors suffer from short-term stability that makes them unsuitable for commercial use. In this regard, a group of factors can be involved, among which the type of electrode used, the length of the aptamer, and the strength of bond between aptamer and electrode surface can be mentioned. Considering that in aptamer-based electrochemical sensors, the most common method of immobilizing the aptamer to the electrode surface is through thiol–gold self-assembly, this kind of biosensor suffers from short-term stability because of the desorption of this chemistry over time and during exposure to dry air, high temperatures, voltage pulsing, and biological fluids. This desorption action results in the simultaneous removal of the sensor moieties and thiol passivation from the electrode surface, preventing them from settling for more than a few hours [69]. However, some researchers could overcome this challenge by keeping electrodes in buffered solutions and dry air to extend shelf-life storage, which is required for successful commercialization. Furthermore, three strategies could increase the stability of thiol self-assembled monolayers including (1) the use of multi-dentate anchoring groups in order to increase the number of attachment points to the electrode surface, (2) the crosslinking of thiol moieties after surface deposition, and (3) the application of hydrophobic thiols to decrease its solubility [69].

Regarding the effect of aptamer length on the shelf-life storage of electrochemical aptasensors, it has been found that long aptamers can be easily detached from the electrode surface than shorter ones [68].

Therefore, in order to develop an electrochemical aptasensor with good performance and applicability, which is the ultimate goal of all this research, the optimization of these parameters is necessary.

## 11. Patents and Commercial Biosensors in Clinical Analysis

Research on electrochemical biosensors is developing rapidly through the innovation and improvement of nanomaterials, bioreceptors, and strategies of detection. Many research products in the field of electrochemical biosensors for the detection of biomarkers have been patented (Table 7). In these patents, different kinds of bioreceptors including enzymes, antibodies, and aptamers have been applied for the detection of various biomarkers.

**Table 1 pharmaceuticals-15-00995-t001:** Recent advances in electrochemical aptasensors for the detection of cancer biomarkers.

Principle of Detection	Biomarker	Aptamer Sequence	LOD	Linear Range	Sample	Features	Ref.
Modification of gold electrode using microgel particles of AuNPs and DPV measurement	miRNA-2	Complementary sequence to miRNA-2	1.35 aM	10 aM–1 pM	Serum	High sensitivity and selectivity, low stability (1 week), no need for sample preparation and purification	[24]
Modification of GCE surface with AuNPs and 3D graphene hydrogel (AuNPs/3DGH), aptamer immobilization and sandwich-type detection	CEA, CA 15-3	CA 15-3 Aptamer: 5′-NH_2_-C6-GCAGTTGATCCTTTGGATACCCTGG-3′	11.2 pg mL^−1^, 11.2 × 10^−2^ U mL^−1^	1.0 × 10^−2^ to 75.0 ng mL^−1^, 1.0 × 10^−2^ to 150.0 U mL^−1^	Serum	High sensitivity and selectivity, satisfactory reproducibility, short-term stability, long incubation time (50 min)	[25]
		CEA Aptamer I: 5′-NH_2_-C6-ATACCAGCTTATTCAATT-3′					
		CEA Aptamer II: 5′-NH_2_-C6-CCCATAGGGAAGTGGGGGA-3′					
Modification of GCE using nanoflower-like MoS_2_ functional interface and signal amplified SiO_2_ nanoprobe and SWV measurement	PSA, sarcosine	nr	2.5 fg mL^−1^, 14.4 fg mL^−1^	1 fg mL^−1^–500 ng mL^−1^, 1 fg mL^−1^–1 µg mL^−1^	Serum	High sensitivity and selectivity, simultaneous detection of two biomarkers, no evaluation of stability, long incubation time (60–120 min)	[27]
Modification of gold electrode using thiolated LC-18 aptamer; CV and non-Faradic EIS measurement	Lung cancer-related proteins and cells	5′-SH-(CH_2_)_6_ CTCCTCTGACTGTAA	nr *	nr	Serum	No evaluation of sensitivity, selectivity and stability of biosensor, short incubation time (30 min)	[29]
		CCACGTGCCCGAACGCGAGTTGAGTTCCGAGAGCTCCGACTTCTTGCATAGGTAGTCCAGAAGCC-3′					
Modification of GCE with a nanocomposite of nickel hexacyanoferrate (NiHCF) nanocubes as the in-situ signal probe and polydopamine functionalized graphene (PDA@Gr) as substrate, and immobilization of CA125 aptamer; DPV measurement	CA125	5′-NH_2_-C_6_H_12_-CTC ACT ATA GGG AGA CAA GAA TAA ACG CTC AA-3′	0.076 pg mL^−1^	0.1 pg mL^−1^–1.0 µg mL^−1^	Serum	High sensitivity and selectivity, short incubation time (20 min), good stability during storage (3 weeks), excellent reproducibility	[32]
Modification of magnetic beads with MUC1 aptamer and hybridization with blocker DNA probe and MUC1 quantification	MUC1	5′-Biotin-TTTTTTGCAGTTGATCCTTTGGATACCCTGG-3′	0.72 pg mL^−1^	5 pg mL^−1^–50 ng mL^−1^	Serum	High sensitivity and selectivity, long detection time (>4 h), no evaluation of stability	[70]
by introducing the released DNA as DNA walkers and Exo III as driven forces on electrode; DPV measurement							
Modification of GCE with graphene quantum dots, polypyrrole and cobalt phthalocyanine and immobilization of HER2 aptamer through amide linkage; EIS measurement	HER2	5′-/5-AmMC6/AAC-CGC-CCA-AAT-CCC-TAA-GAG-TCT-GCA-CTT-GTC-ATT-TTG-TAT-ATG-TAT-TTG GTT-TTT-GGC-TCT-CAC-AGA-CAC-ACT-ACA-CAC-GCA-CA-3′	0.00141 ng mL^−1^	1–10 ng mL^−1^	Serum	High sensitivity and selectivity, short-term stability (4 days), acceptable reproducibility, short incubation time (30 min)	[71]
Immobilization of biotin labeled aptamer on the surface of carboxylated graphene oxide (CGO)/FTO electrode modified with gold platinum bimetallic nanoparticles; DPV measurement	MUC1	5′-Btn GGGAGACAAGAATAAACGCTCAAGCAGTTGATCCTTTGGATACCCTGGTTCGACAGGAGGCTCACAACAGGC-3′	0.79 fM	1 fM–100 nM	Serum	High sensitivity, good selectivity, relatively good stability (15 days), good reproducibility, short incubation time (25 min)	[72]
Physical adsorption of MUC1 aptamer on the surface of needle-shaped disposable 3D-printed electrode and DPV measurement	MUC1	5′-GCAGTTGATCCTTTGGATACCCTGG-3′	80 nM	20–1000 nM	Serum	High sensitivity, good selectivity, short incubation time (30 min), no evaluation of stability	[73]
Modification of screen printed electrode with Au-graphene nanocpmposite as sensing platform, aptamer immobilization; and application of copper sulfide-graphene (CuS-GR) nanocomposite as label; DPV measurement	Acute leukemia cells (CCRF-CEM)	5′-ATCTAACTGCTGCGCCGCCGGGA AAATACTGTACGGTTAGA-3′	18 cell mL^−1^	50 – 1 × 10^6^ cell mL^−1^	Serum	High sensitivity and selectivity, long incubation time (5 h), good stability (21 days), good reproducibility	[74]
Immobilization of aptamer-modified DNA nanotetrahedron coupled with Au on the gold electrode and signal amplification using AuNPs-DNA conjugates coupled with horseradish peroxidase; amperometric measurement	HepG2 liver cancer exosomes	5′-CAC-CCC-ACC-TCG-CTC-CCG-TGA-CAC-TAA-TGC-TA-AAAAAAAAAA-3′	1.66 × 10^4^ particles mL^−1^	2.16 × 10^4^–7.50 × 10^7^ particles mL^−1^	Mice plasma	Good sensitivity and selectivity, short-term stability (48 h), long incubation time (4 h)	[75]
Immobilization of thiol-functionalized aptamer on the surface of SPCE modified with AuNPs/fullerene C_60_-chitosan-ionic liquid/multiwalled carbon nanotubes; EIS and DPV measurement	PSA	5′-HS-(CH_2_)_6_-TTTTTA-ATT-AAA-GCT-CGC-CAT-CAA-ATA-GCT-TT-3′	0.5 pg mL^−1^ (EIS method)	1–200 pg mL^−1^ (EIS method)	Serum	High sensitivity and selectivity, long incubation time (50 min), good stability (1 month), good reproducibility	[76]
			1.5 ng mL^−1^ (DPV method)	2.5–90 ng mL^−1^ (DPV method)			
A displacement electrochemical aptasensor based on amidoxime-modified polyacrylonitrile nanofibers decorated with Ag nanoparticles and aptamer-cDNA duplex and target induced strand displacement; DPV measurement	CA125	5′-NH_2_-TTATCGTACGACAGTCATCCTACAC-3′	0.0042 U mL^−1^	0.01–350 U mL^−1^	Serum	High sensitivity, short incubation time (20 min), short-term stability (10 days), acceptable selectivity, reproducibility, repeatability	[77]
Modification of GCE with 1,3,6,8-tetra(4-carboxylphenyl)pyrene/melamine covalent—organic framework (COF) nanowires as a platform for anchoring ssDNA which was hybridized with the complementary aptamer probes of target; DPV measurement	miRNA 155 and miRNA 122	miRNA-155 aptamer: 5′-NH_2_-AC-CCC-UAU-CAC-GAU-UAG-CAU-UAA-3′	6.7 fM	0.01–1000 pM	Serum	High sensitivity and selectivity, and acceptable recyclability, good reproducibility, acceptable stability (15 days)	[78]
		miRNA-122 aptamer: 5′-NH_2_-C-AAA-CAC-CAU-UGU-CAC-ACU-CCA-3′	1.5 fM				
Modification of gold electrode with capture aptamer, target recognition, addition of second aptamer labeled with silica nanoparticle/CdSe complex; stripping square wave voltammetry (SSWV) measurement	Epithelial cell adhesion molecule (EpCAM)	Capture aptamer: 5′-SH-CACTACAGAGGTTGCGTCTGTCCCACGTTGTCATGGGGGGTTGGCCTG-3′	10 aM	10 aM–100 pM	-	High sensitivity and selectivity, good stability (20 days), long detection time (2 h)	[79]
		Second aptamer: 5′-biotin-CACTACAGAGGTTGCGTCTGTC CCACGTTGTCATGGGGGGTTGGCCTG-3′					

* nr: not reported.

**Table 2 pharmaceuticals-15-00995-t002:** Recent advances in electrochemical aptasensors for the detection of cardiac biomarkers.

Principle of Detection	Biomarker	Aptamer Sequence	LOD	Linear Range	Sample	Features	Ref.
Aptasensor coupling with CRISPR/Cas12a system and MNPs, surface modification of gold electrode with methylene blue-modified nucleic acid probe, DPV measurement	cTnI	5′-CGTGCAGTACGCCAACCTTTCTCATGCGCTGCCCCTCTTATTTTTTTTTT-biotin-3′	10 pg mL^−1^	100–50,000 pg mL^−1^	Serum	High sensitivity and selectivity, high reproducibility, long detection time (60 min), no evaluation of stability during storage.	[35]
Aptamer immobilization on the surface of carboxyethylsilanetriol-modified graphene oxide coated SPCE and addition of analyte and second aptemer conjugated with peroxidase, amperometric measurement	cTnI	5′-NH_2_-(CH2)_6-_CGTGCAGTACGCCAACCTTTCTCATGCGCTGCCCCTCTTA-3′	0.6 pg mL^−1^	1.0 pg mL^−1^ to 1.0 µg mL^−1^	Serum	High sensitivity and selectivity, good reproducibility, good stability during storage (3 weeks), long incubation time (>2 h)	[36]
Aptamer immobilization on the surface of GO modified with poly deep eutectic solvent and AuNPs, EIS measurement	CRP	nr *	0.0003 ng mL^−1^	0.01–50 ng mL^−1^	Serum	High sensitivity and selectivity, short-term stability (10 days), no evaluation of detection time	[37]
GCE modification with polydopamine/AuNPs, antibody immobilization, detection using Mn_3_(PO_4_)_2_/CRP aptamer nanosheets, SWV measurement	CRP	5′-CGAAGGGGATTCGAGGGGTGATTGCGTGCTCCA-TTTGGT-G-3′	0.37 pg mL^−1^	1–1000 pg mL^−1^	Serum	Excellent sensitivity, high selectivity and reproducibility, long incubation time (2 h)	[38]
Modification of gold electrode with ferrocene-based covalent organic framework nanosheets (Fc-COFNs) and aptamer immobilization; release of aptamer from Fc-COFNs surface upon target addition and recovery of electrochemical signal; DPV measurement	cTnI	5’-CGTG-CAGT-ACGC-CAAC-CTTT-CTCA-TGCG CTGC CCCT CTTA-3’	2.6 fg mL^−1^	10 fg mL^−1^–10 ng mL^−1^	Serum	Excellent sensitivity, high selectivity, short-term stability (7 days), good reproducibility	[80]
Aptamer immobilization on the surface of GCE modified with copper nanowires/molybdenum disulfide/reduced graphene oxide (CuNWs/MoS2/rGO); DPV measurement	cTnI	5′-CGT-GCA-GTA-CGC-CAA-CCT-TTC-TCA-TGC-GCT-GCC-CCT-CTT-A-3′	10 × 10^−13^ g mL^−1^	5 × 10^−13^–10 × 10^−10^ g mL^−1^	Serum	High sensitivity and selectivity, relatively short incubation time (40 min), long-term stability (30 days), good repeatability	[81]
Sandwich complex formation between immobilized first aptamer on the surface of SPCE modified with zirconium-carbon loaded with Au (Au/Zr–C), cTnI, and a second aptamer labeled with snowflake-like PtCuNi; amperometric measurement	cTnI	First aptamer: SH-(C6)-5′-CGTGCAGTACGCCAACCTTTCTCATGCGCTGCCCCTCTTA-3′	1.24 × 10^−3^ pg mL^−1^	100 ng mL^−1^–0.01 pg mL^−1^	Serum	High sensitivity, good selectivity, satisfying reproducibility, outstanding stability (21 days), and good recovery, long detection time (100 min)	[82]
		Second aptamer: SH-(C6)-5′-CGCATGCCAAACGTTGCCTCATAGTTCCCTCCCCGTGTCC-3′					
Sandwich complex formation between immobilized first aptamer on the surface of screen-printed gold electrode, cTnI, and second aptamer labeled with Core-shell Pd@Pt dendritic bimetallic nanoparticles loaded on melamine modified hollow mesoporous carbon spheres (Pd@Pt DNs/NH_2_-HMCS) for the reduction of H_2_O_2_, amperometric measurement	cTnI	First aptamer: 5′-SH-(CH_2_)_6_-CGTGCAGTACGCCAACCTTTCTC-ATGCGCTGCCCCTCTTA-3′	15.4 fg mL^−1^	0.1 pg mL^−1^–100 ng mL^−1^	Serum	High sensitivity and selectivity, good stability (3 weeks), good reproducibility, long detection time (100 min)	[83]
		Second aptamer: 5′-SH-(CH_2_)_6_-CGCATGCCAAA-CGTTGCCTCATAGTTCCCTCCCCGTGTCC-3′					
Immobilization of thiol-functionalized DNA aptamer on the surface of fluorine-doped tin oxide (FTO) electrode modified with boron nitride nanosheets and AuNPs; DPV measurement	Myoglobin	5′-CCCTCCTTTCCTTCGACGTAGATCTGCTGCGTTGTTCCGA-3′	34.6 ng mL^−1^	0.1−100 µg mL^−1^	Serum	High sensitivity and selectivity, good stability (30 days), good reproducibility	[84]
Aptamer immobilization on the surface of gold electrode coated with bimetallic MnCo oxide nanohybrids (MnO_x_CoO_y_); EIS measurement	Myoglobin	5′-CCC-TCC-TTT-CCT-TCG-ACG-TAG-ATC-TGC-TGC-GTT-GTT-CCG-A-3′	0.56 fg mL^−1^	0.01–2000 pg mL^−1^	Serum	High sensitivity and selectivity, good stability (15 days), good regenerability and reproducibility	[85]
Immobilization of biotin-linked aptamer on the surface of SPE modified with Cellulose acetate-MoS_2_ nanopetal hybrid; EIS measurement	cTnI	5′-CGT-GCA-GTA-CGC-CAA-CCT-TTC-TCA-TGC-GCT-GCC-CCT-CTT-AAA-AAA-AAA-AAA-AAA-AAA-AAA-AAA-A-3′	10 fM	10 fM–1 nM	Serum	High sensitivity and selectivity, long-term stability (6 weeks), short incubation time (15 min), good reproducibility	[86]
Immobilization of thiol-linked aptamer on the surface of AuNPs modified Ti sheets; DPV measurement	cTnI	5′-CGTGCAGTACGCCAACCTTTCTCATGCGCTGCCCCTCTTA-3′	0.18 pM	1–1100 pM	Serum	High sensitivity and selectivity, simple, low-cost, long incubation time (50 min), short-term stability (7 days), no reusability	[87]

* nr: not reported.

**Table 3 pharmaceuticals-15-00995-t003:** Recent advances in electrochemical aptasensors for the detection of Alzheimer’s disease biomarkers.

Principle of Detection	Biomarker	Aptamer Sequence	LOD	Linear Range	Sample	Features	Ref.
Immobilization of the thiol-modified amyloid-β DNA aptamers on the CFP/AuPt electrode and DPV measurement	Amyloid-β	5′-SH-(CH_2_)_6_-GCTGC-CTGTGGTGTTGGGGC GGGTG CG-3′	0.16 pg mL^−1^ in buffer, 0.9 pg mL^−1^ in serum	0.5–10,000 pg mL^−1^	Serum	High sensitivity, good selectivity, good stability (60 days), relatively short detection time (1 h)	[43]
Ce Aptamer immobilization on the surface of AuNPs/MoSe_2_ nanosheets modified electrode and photoelectrochemical detection	Tau-381 protein	5′-SH-GCGGAGCGTGGCAGG-3′	0.3 fM	0.5 fM to 1.0 nM	Serum	High sensitivity and selectivity, short-term stability (28 days), long detection time (>2 h)	[44]
Aptamer immobilization on the surface of GCE modified with hyaluronic acid functionalized polydopamine	Thrombin	5′-NH_2_-(CH_2_)_6_-AGTCCGTGGTAGGGCAGGTTGGGGTGACT-3′	0.03 pM	0.1 pM to 1.0 nM	Serum	High sensitivity and selectivity, good reproducibility, short detection time (40 min), short-term stability (15 days)	[45]
Immobilization of thiol-terminated ssDNA aptamer on the surface of gold electrode through Au-S interactions; EIS measurement	Amyloid-β	5′-OH-(CH_2_)_6_-S-S-(CH_2_)_6_-GCCTGTGGTGTTGGGGCGGGTGCG-3′	0.03 nM	0.1–500 nM	-	High sensitivity and selectivity, easy fabrication, relatively short incubation time (30 min), short-term stability (14 days)	[88]
Immobilization of complementary sequence of aptamer on the surface of SPCE and application of exonuclease I (Exo I), terminal deoxynucleotidyl transferase (TdT) and methylene blue as sensing platform; DPV measurement	α-synuclein	5′-TTTTTGGTGGCTGGAGGGGGCGCGAACG-3′	10 pM	60 pM–150 nM	Serum	High sensitivity and selectivity, good stability (15 days), high repeatability, long detection time	[89]
Immobilization of aptamer 1 on the surface of gold electrode, deposition of aptamer 2 and Aβ oligomers on the electrode surface, triggering of hybridization chain reaction (HCR) by aptamer 2, AgNPs adsorption on the electrode surface; linear sweep stripping voltammetry (LSV) measurement	Amyloid-β	Aptamer 1: 5′-AAAAAAAAAAGAGAGCCTGTGTTGGGGCGGGTGCG-3′	430 fM	1 pM–10 nM	Serum	High sensitivity and selectivity, long detection time (>2 h), short-term stability (14 days), excellent reproducibility	[90]
		Aptamer 2: 5′-AGAGAGCCTGTGTTGGGGCGGGTGCGGTTATTAATGTGTGATGT-3′					
Modification of the gold screen-printed electrode with the hemin-aptamer conjugate, binding of thrombin to aptamer, triggering of the aptamer folding into the hemin-G-quadruplex DNAzyme structure and electrocatalytic activity; CV measurement	Thrombin	5′-SH-C6-AGT-CCG-TGG-TAG-GGC-AGG-TTG-GGG-TGA-CTT-TTT-TTT-TTT-C7-NH2-3′	0.5 fM	1 fM–100 fM	Serum	High sensitivity and selectivity, short-term stability (4–5 days), long detection time	[91]
Aptamer immobilization on the surface of gold electrode, complex formation between aptamer and target coordinated with Cu^2+^, electrochemiluminescence signal production via a catalytic reaction between Cu^2+^-Aβ-aptamers and the dissolved O_2_	Amyloid-β	5′-HS-GCC-TGT-GGT-GTT-GGG-GCG-GGT-GCG-3′	3.5 × 10^−14^ M	10 × 10^−13^ M–10 × 10^−10^ M	Serum	High sensitivity and selectivity, acceptable reproducibility, no evaluation of storage stability, long detection time	[92]

**Table 4 pharmaceuticals-15-00995-t004:** Recent advances in electrochemical aptasensors for the detection of MS biomarkers.

Principle of Detection	Biomarker	Aptamer Sequence	LOD	Linear Range	Sample	Features	Ref.
Aptamer immobilization on the surface of graphite sheet coated with a nanocomposite of SWCNT and PPY; DPV measurement	miR-155	5′-NH2-ACCCCUAUCACG-AUUAGCAUUAA-3′	10 aM	10 aM to 1 µM	Serum	High sensitivity and selectivity, good reproducibility, long incubation time (2 h), no evaluation of stability	[46]
Aptamer immobilization on the surface of gold electrode modified with AuNPs; EIS measurement	IL-17RA	5′-thiol-CTTGGATCACCATAGTCGCTAGTCGAGGCT-3′	2.13 pg mL^−1^	10–10,000 pg mL^−1^	Serum	High sensitivity and selectivity, high reproducibility, short incubation time (30 min), no evaluation of stability	[49]
Aptamer immobilization on the surface of gold electrode and application of alternating current electroosmotic (ACEO) flow phenomenon for the enhanced target hybridization of microRNA-155; EIS measurement	miR-155	5′-SH-AAA-AAA-AAC-CCC-UAU-CAC-GAU-UAG-CAU-UAA-3′	1 aM	1 aM–10 pM	Serum	High sensitivity and selectivity, relatively short incubation time (1 h), no evaluation of stability and reproducibility	[93]
A signal on-off ratiometric electrochemical aptasensor based on aptamer immobilization on the surface of GCE modified with AuNPS-MXene, hairpin probe labeled with ferrocene as signal probe, aptamer labeled with methyl blue; Alternating current voltammetry (ACV) measurement	Thrombin	nr *	1.67 fM	5.0 fM–1.0 pM	Serum	High sensitivity and selectivity, short-term stability (10 days), satisfactory reproducibility	[94]
Sandwich-type thrombin aptasensor based on Ag nanowires & particles electrode and signal amplification of Pt/ZnFe_2_O_4_; amperometric measurement	Thrombin	Aptamer 1: 5′-NH_2_-(CH_2_)_6_-GGT-TGG-TGT-GGT-TGG-3′	0.016 pM	0.05 pM–35 nM	Serum	High sensitivity and selectivity, satisfactory reproducibility, good stability (21 days)	[95]
		Aptamer 2: 5′-SH-(CH_2_)_6_-AGT-CCGTGG-TAG-GGC-AGG-TTG-GGG-TGA-CT-3′					

* nr: not reported.

**Table 5 pharmaceuticals-15-00995-t005:** Recent advances in electrochemical aptasensors for the detection of malaria biomarkers.

Principle of Detection	Biomarker	Aptamer Sequence	LOD	Linear Range	Sample	Features	Ref.
Aptamer immobilization on the surface of gold electrode and coating with PEG, EIS measurement	PfLDH	5′-HO-(CH_2_)_6_-S-S-(CH_2_)_6_-O-CTGGGCGGTAGAACCATA-GTGACCCAGCCGTCTAC-3′	1.49 pM	4.5 pM–100.0 nM	Serum	High sensitivity and selectivity, short incubation time (45 min), short-term stability of PEG layer (1 day)	[51]
Immobilization of thiolated aptamer on the surface of gold electrode, EIS measurement	PfGDH	5′-SH-(CH_2_)_6_-TTT-TCA-CCT-CAT-ACG-ACT-CAC-TAT-AGC-GGA-TCC-GAG-CCG-GGG-TGT-TCT-GTT-GGC-GGG-GGC-GGT-GGG-CGG-GCT-GGC-TCG-AAC-AAG-CTT-GC-3′	0.77 pM	100 fM–100 nM	Serum	High sensitivity and selectivity, short incubation time (30 min), no evaluation of stability	[52]
Immobilization of amine-functionalized aptamer on the surface of gold electrode, EIS measurement	HRP-II	5′CACCTAATACGACTCACTATAGCGGATCC-GA-N40-CTGGCTCGAACAAGCTTGC-3′	3.15 pM	1–500 pM	Serum	High sensitivity and selectivity, no evaluation of stability	[53]
System-integrated two-dimensional field-effect transistors (2DBioFETs) of reduced graphene oxide (rGO) as transducer	PfLDH	5′-CTGGGCGGTAGAACCATA-GTGACCCAGCCGTCTAC-3′	0.78 fM	0.78 fM–100 nM	Serum	High sensitivity and selectivity, no evaluation of stability and reproducibility	[96]
Immobilization of aptamer modified with a methylene blue (MB) reporter on a gold sensor surface for square-wave voltammetry interrogation	PfHRP2	5′-thiol-GCTTATCCGATGCAGACCCCTTCGGTCCTGCCCTC-MB-3′	3.73 nM		Serum	High sensitivity and selectivity, good stability (14 days)	[97]

**Table 6 pharmaceuticals-15-00995-t006:** Recent advances in electrochemical aptasensors for the detection of diabetes biomarkers.

Principle of Detection	Biomarker	Aptamer Sequence	LOD	Linear Range	Sample	Features	Ref.
Aptamer immobilization on the surface of SPCE modified with AuNPs; SWV measurement	HbA1c and tHb	5′-GGGGACACAGCAACAC	0.2 and 0.34 ng mL^−1^	100 pg mL^−1^ 10 µg mL^−1^	Whole blood	High sensitivity and selectivity, short incubation time (30 min), no need for sample pretreatment, no evaluation of stability	[58]
		ACCCACCCACCAGCCCCAGCATCATGCCCATCCGTCGTGTGTG-3′					
		5′-ACGCACACCAGAGACA					
		AGTAGCCCCCCAAACGCGGCCACGGAACGCAGCACCTCCATGGC-3′					
Aptamer immobilization on the surface of SPCE modified with streptavidin; SWV measurement	GHSA and HSA	5′-TGCGGTTGTAGTACTCG	3 ng mL^−1^ and 0.2 µg mL^−1^	2 × 10^−6^–16 mg mL^−1^ and 5 × 10^−5^–100 mg mL^−1^	Serum	High sensitivity and selectivity, short incubation time (40 min), relatively long-term stability (4 weeks), good reproducibility	[59]
		TGGCCG-3′					
		5′-ATACCAGCTTATTCAATTCCCCCGGCTTTGGTTTAGAGGTAGTTGCTCATTACTTGTACG CTCCGGATGAGATAGTAAGTGCAATCT-3′					
Aptamer immobilization on the surface of SPCE modified with flower-like gold microstructures; DPV measurement	Serpin A12	5′-Thiol-C6-ATACCAGCTTATTCAATTGGGCGGTGGGGGGGGTAGTGGGTGTTATGGCGATCGTGGAGATAGTAAGTGCAATCT-3′	0.02 ng mL^−1^	0.039–10 ng mL^−1^	Serum	High sensitivity and selectivity, short incubation time (30 min), short-term stability (2 weeks), good reproducibility	[60]
			0.031 ng mL^−1^				
Aptamer immobilization on the surface of gold electrode and utilization of hybridization chain reaction (HCR) and CeO_2_ nanoparticles as a cascade signal amplification strategy; DPV measurement	VEGF	5′-ACTCTTGTCTGGAAGACG	7.39 fg mL^−1^	10–10^5^ fg mL^−1^	Tear	Ultrahigh sensitivity and high selectivity, short-term stability (5 days), acceptable reproducibility, long detection time	[63]
		GAAACCCTGCACTCCCGTCTTCCAGACAAGAGTGCAGGG-3′					
Immobilization of Methylene blue (MB)-modified insulin-binding aptamer as “signal-off” probe and ptamer/Ferrocene (Fc) co-modified AuNPs as the “signal-on” probe on the surface of gold electrode; SWV measurement	Insulin	Insulin binding aptamer: 5′-CCA-CCA-CCC-GGG-GGT-CCT-AGG-GTC-AAC-AAA-MB-3′	0.1 pM	10 pM–10 nM	Serum	High sensitivity and selectivity, long incubation time (3 h), short-term stability (1 week), good reproducibility	[98]
		MB-modified aptamer: 5′-GGT-GGT-GGG-GGG-GGT-GGT-AGG-GTG-TCT-TCT-MB-3′					
Immobilization of thiolated aptamers terminated with redox probes methylene blue on the surface of SPCE modified with AuNPs; SWV measurement	Glucose	Glucose aptamer: 5′–HS–HS-C6-CTCTCGGGACGACCGTGTGTGTTGCTCTGTAACAGTGTCCATTGTCGTCCC-MB-3’	0.08 mM	0.1–50 mM	Saliva	High sensitivity and selectivity, relatively short incubation time (30–45 min), long incubation time (30 days), equipped with smartphone signal readout	[99]
	Insulin	Insulin aptamer: 5′-HS-HS-C6-AAAAGGTGGTGGGGGGGGTTGGTAGGGTGTCTTCT-MB-3′	0.85 nM	0.05–15 nM			
Coating of functionalized mesoporous silica thin-film on the electrode, aptamer hybridization with the cDNA immobilized on the silica film in order to cap the mesochannels, triggering of insulin the opening of mesochannels to regulate the controlled diffusion of Fe(CN)_6_^3−/4−^(CN)_6_-; DPV measurement	Insulin	5′-GGT-GGT-GGG-GGG-GGT-TGG-TAG-GGT-GTC-TTC-3′	3.0 nM	10.0–350.0 nM	Serum	High sensitivity and selectivity, good reproducibility, no evaluation of stability	[100]

**Table 7 pharmaceuticals-15-00995-t007:** Selected patents and marketed products related to the electrochemical biosensors for biomarker detection.

Patent Number/Brand of Marketed Product	Publication Date	Title/Strategy	Biomarker	Bioreceptor	Ref.
US20160331235A1	17 November 2016	System and method for measuring biological fluid biomarkers	Small molecules, proteins, metabolites, and/or electrolytes in sweat	nr	[101]
EP20200382721	9 February 2022	Biosensor system for multiplexed detection of biomarkers	Biomarkers of Chronic Obstructive Pulmonary Disease (COPD): tumor necrosis factor alpha (TNF-α), cytokine interleukin-8 (IL-8), Myeloperoxidase (MPO)	Antibody or DNA strand	[102]
US20150247816	9 March 2015	Label-free electrochemical biosensor	Saliva cortisol and other biomolecules	Binding protein, antibody, aptamer	[103]
US20190317089	17 October 2019	Multi-array impedimetric biosensors for the detection of concussion and traumatic brain injuries	Biomarkers related to brain injury (Tau proteins, Glial Fibrilar Acidic Protein (GFAP) and Ubiquitin C-Terminal Hydrolase L1 (UCH-L1))	Antibody, aptamer	[104]
US11166653	9 November 2021	Reconfigurable, multi-technique electrochemical portable biosensor	Glucose, lactoferrin	Antibody, enzyme	[105]
US20180136190	17 May 2018	Biosensors for detecting cholesterol and OxLDL in blood sample	Cholesterol and OxLDL	Enzyme	[106]
US11045806	29 June 2021	Integrated type microfluidic electrochemical biosensor system and method for rapid biochemical analysis	PSA and human liver cancer marker AFP	Antibody	[107]
US20180292400	11 October 2018	Development and parameter assessment for vertically aligned platinum wire aptasensor arrays for the impedimetric detection of cardiac biomarkers	Cardiac biomarkers (brain natriuretic peptide (BNP) and TroponinT (TnT))	Aptamer	[108]
US10107824	23 October 2018	Method for detecting cardiovascular disease biomarker	Cardiac biomarkers (Troponin I and NT-proBNP)	Antibody, aptamer	[109]
US20210338157	4 November 2021	Pacifier sensor for biomarker monitoring	Saliva biomarkers (e.g. glucose, glucose, lactate, uric acid, cortisol, etc.)	Enzyme	[110]
Accu-Chek Advantage	-	Electrochemical device with palladium electrode	Glucose	Glucose oxidase	-
Accu-Chek Advantage	-	Electrochemical device with palladium electrode	Glucose	Glucose dehydrogenase	-
Accu-Chek Aviva	-	Electrochemical device with gold electrode	Glucose	Glucose dehydrogenase	-
One Touch Ultra	-	Electrochemical device with carbon electrode	Glucose	Glucose oxidase	-
The edge (ApexBio, Hsinchu, Taiwan)		Enzymatic electrochemical device	Lactate	Lactate oxidase	
The edge (ApexBio, Hsinchu, Taiwan)		Enzymatic electrochemical device	Uric acid	Enzyme	-
Q. STEPS G/C ADMS (American Screening Corp., Shreveport, LA, USA)		Enzymatic electrochemical device	Glucose and cholesterol	Enzyme	-
CardioChek (Pts Diagnostics, Changsha, China)		Enzymatic electrochemical device	Cholesterol	Enzyme	-
MultiSure (ApexBio, Hsinchu, Taiwan)		Enzymatic electrochemical device	Glucose and uric acid	Enzyme	-

Some commercial electrochemical devices for biomarker detection have been listed in Table 7. As can be observed, despite the large number of studies conducted in the field of electrochemical biosensors, only a limited number of these biosensors for the detection of glucose, lactate, uric acid, and cholesterol have been commercialized and have entered the market. For other biomarkers, such as cardiac biomarkers, rapid non-electrochemical (i.e., colorimetric) diagnostic tests are available. Regarding the progress of on-chip and microfluidic-based devices in clinical diagnostics, miniaturized electrochemical biosensors can be designed and developed as new versatile platforms for the future detection of different biomarkers. In this regard, some parameters such as low cost, high sensitivity, fast response, and easy operation for analyte detection must be considered. Microfluidic chip design is a critical point that further improves the analytical performances of LOC platforms, especially by reducing the reagent consumption as well as by shortening assay preparation and analysis times. Moreover, these biosensors should not require expensive instrumentation and specialized personnel.

## 12. Conclusions

This review summarized recent developments in electrochemical aptasensors for the diagnosis of biomarkers, which is an attractive field in modern medical diagnostics. The design and construction of biosensors can play an important role in the early diagnosis of many diseases using their specific biomarkers. In this regard, electrochemical aptasensors have received much attention due to their high sensitivity and accuracy, as well as their low cost in terms of the type of aptamer diagnostic element. The review of the literature showed that most of the research in the field of biomarker detection using electrochemical aptasensors has been conducted in relation to the biomarkers of different types of cancer and then heart diseases. Therefore, great efforts are still needed for the development of electrochemical aptasensors for the diagnosis of other biomarkers.

One of the most important goals of the research conducted in the field of biosensors including biomarker aptasensors is their entry into the market for community use. However, there is still a long way to the applicability of biosensors glucose biosensors due to several challenges. Some of the challenges ahead include the short-term stability, long-term incubation with the analyte, and low reproducibility of the developed aptasensors. On the other hand, fabricating an aptasensor with high specificity requires finding specific biomarkers of a disease and also selecting specific aptamers with these biomarkers. Despite many efforts, there are still no specific biomarkers for some diseases such as MS. In this regard, the simultaneous identification of several general biomarkers may be a solution that also requires the construction of biosensors that are able to simultaneously detect several biomarker analytes. To sum up, regarding the recent advances in the field of aptamer-based electrochemical biosensors, great progress is expected in parallel with lab-on-a chip devices and miniaturized platforms such as microfluidics aptasensors with potential clinical diagnostic applications.

## Figures and Tables

**Figure 1 pharmaceuticals-15-00995-f001:**
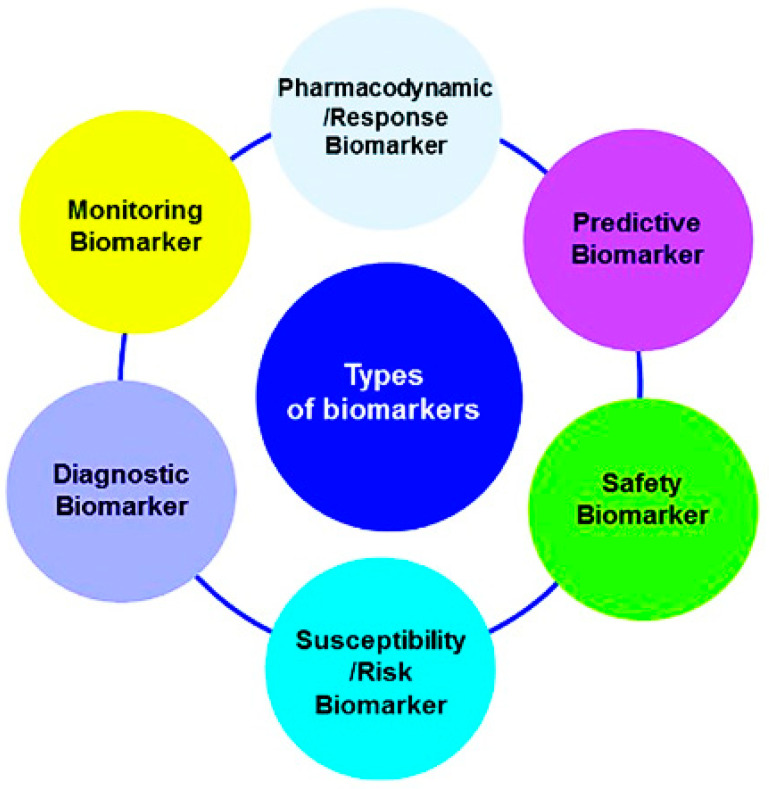
Classification of biomarkers according to their clinical applications. Reprinted from [3].

**Figure 2 pharmaceuticals-15-00995-f002:**
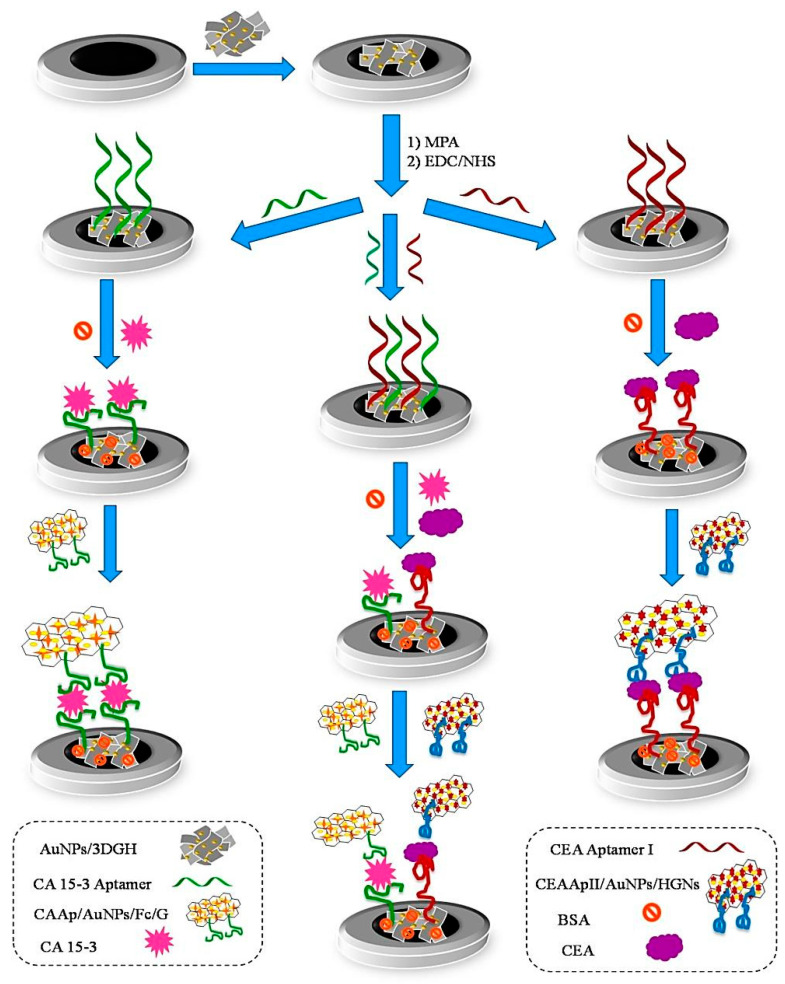
Schematic illustration of the sandwich-type aptasensor for the individual and simultaneous detection of CEA and CA 15-3 breast cancer biomarkers. Reprinted from [25] with permission.

**Figure 3 pharmaceuticals-15-00995-f003:**
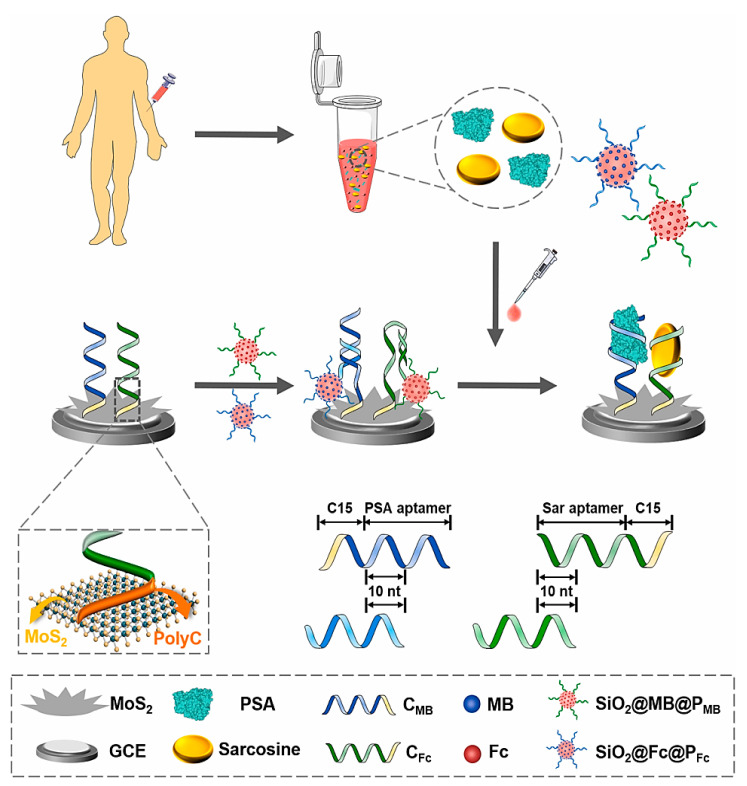
Schematic representation of electrochemical aptasensors for the one-step, simultaneous detection of two prostate cancer biomarkers using nanoflower-like MoS_2_ functional interfaces and signal-amplified SiO_2_ nanoprobes. Reprinted from [27] with permission.

**Figure 4 pharmaceuticals-15-00995-f004:**
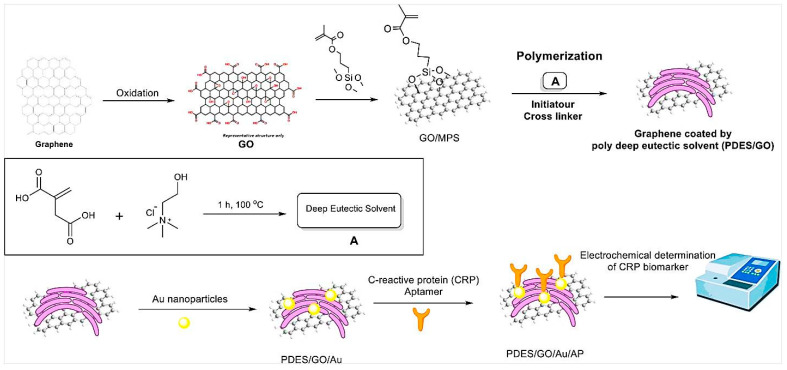
Schematic representation of the fabrication steps for the CRP aptasensor based on GO/PDES/AuNPs. Reprinted from Ref. [37] with permission.

**Figure 5 pharmaceuticals-15-00995-f005:**
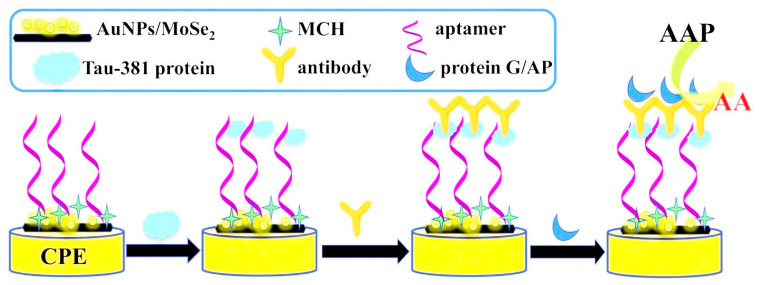
Schematic representation of an enzyme-linked aptamer photoelectrochemical aptasensor for Tau-381 protein. Reprinted from [44] with permission.

**Figure 6 pharmaceuticals-15-00995-f006:**
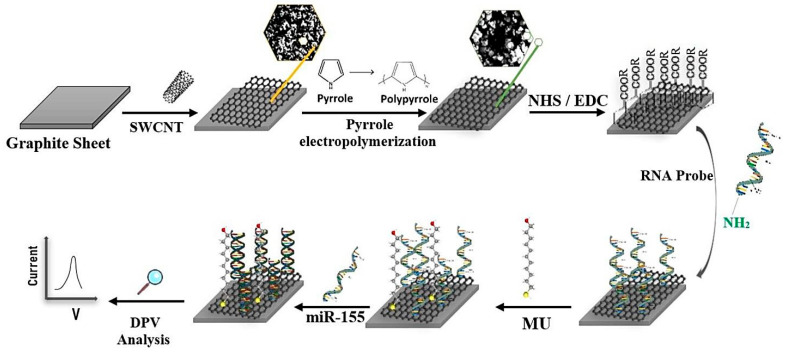
Schematic representation of an electrochemical aptasensor based on the SWCNTs/PPY for miR-155. Reprinted from [46], with permission.

**Figure 7 pharmaceuticals-15-00995-f007:**
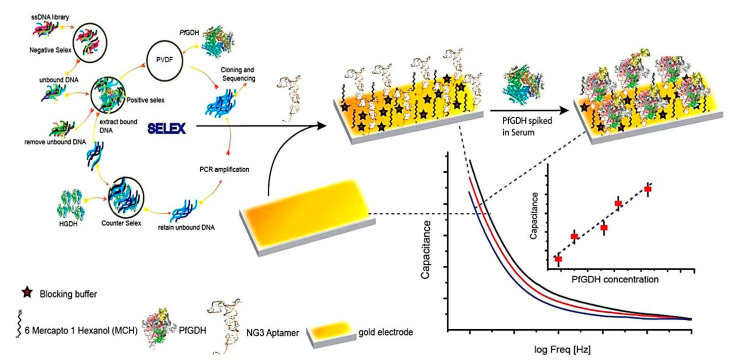
Schematic illustration of apatamer selection and the fabrication steps of a label-free electrochemical aptasensor for the detection of *Pf*GDH. Reprinted from [52] with permission.

**Figure 8 pharmaceuticals-15-00995-f008:**
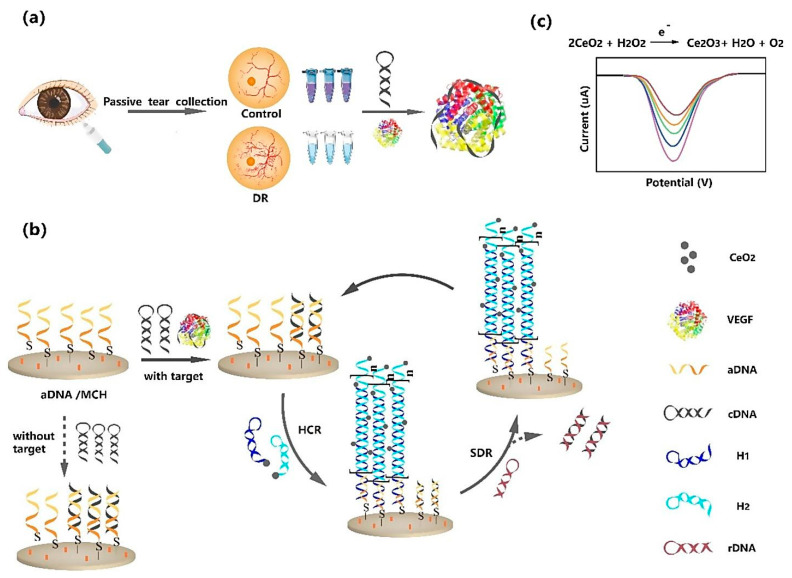
Schematic representation of a VEGF aptasensor with isothermal signal amplification strategy. (**a**) Capture of VEGF; (**b**) signal amplification strategy; (**c**) signal production. Reprinted from [63], with permission.

## Data Availability

Data is contained within the article.

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
