# Peer review of "Recent Advances in Electrochemical Aptasensors for Detection of Biomarkers"

_pharmaceuticals, 2022, doi:10.3390/ph15080995_

Round 1

Reviewer 1 Report

In this manuscript authors have reported the recent advances in electrochemical aptasensors for detection of biomarkers but the following correction is needed.

1. The current manuscript has totally omitted the methodology. This section should also describe the following points

  How many manuscripts were screened?

  How many manuscripts were considered for this study?

  What were the limitations of the methodology used?

3. Research gap should be provided in the manuscript

4. This manuscript should have a table which should include a patented, marketed and innovative electrochemical sensor used for the detection of different diseases.

4. Authors should have gone through this articles which  shown the uses of aptasensor in cancer diagnosis https://doi.org/10.1016/j.matlet.2021.131240.

2. Sensor have been also used for detection of  viral and diagnosis of other diseases. Please discuss in sperate heading or in miscellaneous section.

Author Response

Reviewer 1:

In this manuscript authors have reported the recent advances in electrochemical aptasensors for detection of biomarkers but the following correction is needed.

  1. The current manuscript has totally omitted the methodology. This section should also describe the following points

 Ø  How many manuscripts were screened?

Ø  How many manuscripts were considered for this study?

Ø  What were the limitations of the methodology used?

Response: Thank you very much for your comment. Because this review is not a systematic review, the methodology section has been omitted. However, due to valuable comment of the respected reviewer, some explanations related to the mentioned questions was included in the manuscript (lines 115-123).  

  1. Research gap should be provided in the manuscript

Response: Thank you very much for your comment. The research gap has been discussed in conclusion section lines 744 to 755.

  1. This manuscript should have a table which should include a patented, marketed and innovative electrochemical sensor used for the detection of different diseases.

Response: Thank you very much for your comment. Table 7 was included for patented, marketed and innovative electrochemical sensor used for the detection of different diseases. Moreover, some explanations related to this issue and table was included in lines 713-731.

  1. Authors should have gone through this articles which shown the uses of aptasensor in cancer diagnosis https://doi.org/10.1016/j.matlet.2021.131240.

Response: Thank you very much for your comment. The reference was included (lines 127-130)

  1. Sensor have been also used for detection of viral and diagnosis of other diseases. Please discuss in sperate heading or in miscellaneous section.

Response: Thank you very much for your comment. You are right. A new section was included in the manuscript to provide some information related to the potential of electrochemical aptasensor for detecting other biomarkers as well as microbial pathogens (lines 609-667).

Reviewer 2 Report

Developing sensitive and selective sensors for the detection of biomarkers for various diseases is a very important task in molecular diagnostics. There are several types of biosensors for this purpose based on different recognizing elements, including antibodies, aptamers etc. Among this variety aptasensors are very promising tools for biomarker diagnostic systems. The manuscript "Recent advances in electrochemical aptasensors for detection of biomarkers" summarizes the recent progress in the field of aptasensors development recruiting electrochemical type of analytical signal. The manuscript in very well structurized, and has all necessary references and figures. I totally recommend this manuscript for publication.

As a little comment, I have noticed several spelling mistakes so I would like to recommend additional spell checking. 

Author Response

Reviewer 2:

Developing sensitive and selective sensors for the detection of biomarkers for various diseases is a very important task in molecular diagnostics. There are several types of biosensors for this purpose based on different recognizing elements, including antibodies, aptamers etc. Among this variety aptasensors are very promising tools for biomarker diagnostic systems. The manuscript "Recent advances in electrochemical aptasensors for detection of biomarkers" summarizes the recent progress in the field of aptasensors development recruiting electrochemical type of analytical signal. The manuscript in very well structurized, and has all necessary references and figures. I totally recommend this manuscript for publication.

As a little comment, I have noticed several spelling mistakes so I would like to recommend additional spell checking. 

Response: Thank you very much for your valuable comment. Spell checking was done and required revisions were performed.

Reviewer 3 Report

The manuscript focuses on the review of the recent achievements concerning the use of aptamer-based monolayers for detection of biomarkers of cardiac, cancer, diabietes on other types of diseases. I find the proposed manuscript quite interesting however there is a couple of issues I would like to point out concerning the proposed manuscript:

1) there are a couple of mispellings in the article so the authors should check the manuscript and correct it

2) there should be an unification of units in terms of working parameters such as lower limit of detection and range of linear response

3) in terms of some of the examples of aptasensors such as LC-18 the working parameters should be included in the text and not only in the table

4) I would suggest the authors to include a list of abbreviations of the terms used in the text

5) Future perspectives on possible directions like miniaturization and use of microfluidic systems should be included

6) in my opinion the tables included in the manuscript should be reorganized and divided into categories: the construction of aptasensor and the mode of analytical signal generation; I would also suggest the authors to be more specific in terms of sensitivity and selectivity. The authors should also unify the units of the described aptasensors

Author Response

Reviewer 3:

The manuscript focuses on the review of the recent achievements concerning the use of aptamer-based monolayers for detection of biomarkers of cardiac, cancer, diabietes on other types of diseases. I find the proposed manuscript quite interesting however there is a couple of issues I would like to point out concerning the proposed manuscript:

1) there are a couple of mispellings in the article so the authors should check the manuscript and correct it

Response: Thank you very much for your precision. The manuscript was checked and revised for misspelling.

2) there should be an unification of units in terms of working parameters such as lower limit of detection and range of linear response

Response: Thank you very much for your valuable comment. You are right. However, it is not possible to do this change due to the variety and large number of analytes. Also, I think it is better to state the same unit mentioned in the original article.

3) in terms of some of the examples of aptasensors such as LC-18 the working parameters should be included in the text and not only in the table

Response: Thank you very much for your valuable comment. Working parameters were included in the main text (lines 245-246).

4) I would suggest the authors to include a list of abbreviations of the terms used in the text

Response: a list of abbreviations was provided.

5) Future perspectives on possible directions like miniaturization and use of microfluidic systems should be included

Response: Thank you very much for your valuable comment. Some explanations regarding to miniaturization and microfluidic systems were included in the text (Lines 724-731).

6) in my opinion the tables included in the manuscript should be reorganized and divided into categories: the construction of aptasensor and the mode of analytical signal generation; I would also suggest the authors to be more specific in terms of sensitivity and selectivity. The authors should also unify the units of the described aptasensors

Response: Thank you very much for your comment. However, I think tables in the present format are very clear. Therefore, it is better to remain unchanged. Regarding unit of LOD, as I explained in the previous comment, it is not possible to do this change due to the variety and large number of analytes. Also, I think it is better to state the same unit mentioned in the original article. Moreover, we tried to explain selectivity of the aptasensor according to the author mentions in their articles.

Reviewer 4 Report

The manuscript reviewing recent advances in electrochemical aptasensors for detection of biomarkers by Majdinasab and Marty can provide a useful overview for the readers in this field to refer. I also noted that a majority of the references are within 3-5 recent years, which is concise and informative. Herein, I suggest complementing it with some major points to improve its quality.

1. A section/paragraph briefly introducing and summarizing fundamentals of electrochemical biosensors (working principles, materials, evolution progress, etc.) can be useful and more convenient for the readers to comprehend the content of the manuscript.

2. Section 2: The electrochemical aptasensors have been divided into 3 groups based on the targets (protein, cells, and exosomes). Should nucleic acids (microRNAs) be counted as another group?

3. Section 2, the first paragraph, the last sentence (page 5): As far as I know, PSA is currently not the only biomarker in clinical use for cancer diagnosis (AFP, CA 125, CA 15-3, etc.). This sentence also partially conflicts with the prior one, claiming that a handful of them have been used clinically in the last 30 years. Therefore, please re-check the information and revise if necessary.

Author Response

Reviewer 4:

The manuscript reviewing recent advances in electrochemical aptasensors for detection of biomarkers by Majdinasab and Marty can provide a useful overview for the readers in this field to refer. I also noted that a majority of the references are within 3-5 recent years, which is concise and informative. Herein, I suggest complementing it with some major points to improve its quality.

  1. A section/paragraph briefly introducing and summarizing fundamentals of electrochemical biosensors (working principles, materials, evolution progress, etc.) can be useful and more convenient for the readers to comprehend the content of the manuscript.

Response: Thank you very much for your valuable comment. Some explanations regarding electrochemical biosensors was provided in the “Introduction” section (lines 77-86).

  1. Section 2: The electrochemical aptasensors have been divided into 3 groups based on the targets (protein, cells, and exosomes). Should nucleic acids (microRNAs) be counted as another group?

Response: Thank you very much for your precision and valuable comment  

MicroRNAs (miRNAs) are a class of non-coding RNAs that play important roles in regulating gene expression. miRNAs can be secreted into extracellular fluids and transported to target cells via vesicles, such as exosomes, or by binding to proteins, including Argonautes. Extracellular miRNAs function as chemical messengers to mediate cell-cell communication (https://doi.org/10.3389/fendo.2018.00402).

Increased levels of circulating nucleic acids (DNA, mRNA and microRNA (miRNA)) in the blood reflect pathological processes, including malignant and benign lesions (https://doi.org/10.1038/nrc3066) and (doi: 10.3390/ncrna3010009)

Therefore, microRNAs and circulating nucleic acids as cancer biomarkers have been embedded into exosomes and shouldn’t classify in a separate group.

  1. Section 2, the first paragraph, the last sentence (page 5): As far as I know, PSA is currently not the only biomarker in clinical use for cancer diagnosis (AFP, CA 125, CA 15-3, etc.). This sentence also partially conflicts with the prior one, claiming that a handful of them have been used clinically in the last 30 years. Therefore, please re-check the information and revise if necessary.

Response: Thank you very much for your precision. You are right. I apologize for this mishap. The sentence was revised and used in another section (lines 200-202)

Round 2

Reviewer 1 Report

Thank you Comments incorporated